# Optimal set of leaf and aboveground tree elements for predicting forest functioning

Écio Souza Diniz[1], Eladio Rodríguez Penedo[1], Roger Grau-Andrés [1], Jordi Vayreda[1], Marcos Fernández-Martínez[1],

[1] CREAF, Centre de Recerca Ecològica i Aplicacions Forestals, Cerdanyola del Vallès, 08193 Barcelona, Catalonia, Spain

*Correspondence to*: Écio Souza Diniz (eciodiniz@gmail.com) and Marcos Fernández-Martínez (m.fernandez@creaf.cat)

**Abstract.** The role played by environmental factors in the functioning of forest ecosystems is relatively well known. However, the potential of the elemental composition of trees (i.e., elementomes) as a predictor of forest functioning remains elusive. We assessed the predictive power of elemental composition from different perspectives: testing whether aboveground element stocks or concentrations explain forest production and productivity (i.e., production per unit of standing biomass) better than leaf elements or environmental factors; identifying the optimal set (combination and quantity) of elements that best predicts forest functioning. To do so, we used a forest inventory of 2000 plots in the northeast of the Iberian Peninsula, containing in-site information about the elementomes (C, Ca, K, Mg, N, Na, P, and S) of leaves, branches, stems and barks, in addition to annual biomass production per organ. We found that models using leaf element stocks as predictors achieve the highest explained variation in forest production. The optimal dimensionality was achieved by combining the foliar stocks of C, Ca, K, Mg, N, P, and interactions (C×N, C×P, and N×P). Forest biomass productivity was best predicted by forest age. Hence, our results indicate that leaf element stocks are better predictors of forest biomass production than aboveground element concentrations or stocks, thus hinting toward leaf measurements as critical factors for predicting variations in forest biomass production.

## 1 Introduction

Environmental conditions influence the assembly of tree communities, thus forming different forest types across distinct environmental gradients, e.g., climate and soil variation (Chu et al., 2019; Sardans et al., 2016). Soil nutrient availability (e.g., N, P, K) directly affects tree growth and is thus a key regulator of global forest productivity and forest biomass accumulation (Batjes, 1996; Wiesmeier et al. 2019). The stocks of soil nutrients are influenced by the climatic conditions that drive water availability, temperature-dependent nutrient cycling, and soil organic matter decomposition rates (Zhang et al. 2018c; Mensah et al., 2023). Such environmental conditions encompass specific niches (e.g., climatic and soil conditions) and then drive drive functional adaptations of the species (e.g., morphology or physiology traits) (Lavorel et al., 2007; Augusto et al. 2017; Wang et al., 2022). As the backbone of functional adaptations to such niches, the concentration of elements (e.g., C, N, and P, amongst others) in organisms is a key factor driving ecosystem structure and functioning

(Fernández-Martínez, 2022; Peñuelas et al., 2019). Element concentrations in tree biomass vary along environmental gradients, species, and forest age, which are key drivers of forest functioning (Santiago et al., 2004; Sardans and Peñuelas, 2014). Therefore, investigating the combination and concentration of distinct elements is vital to better understanding forest functioning (e.g., biomass production).

The multi-dimensional concentration of elements of an organism has been defined as the elementome (Peñuelas et al., 2019). Assessing the elementomes of different species allows for a better understanding of how they withstand contrasting environmental conditions since their ecological strategies rely on different element concentrations and functional traits (Peñuelas et al., 2019; Fernández-Martínez, 2022; Reich and Oleksyn, 2004). Within plant elementomes, the importance of the concentrations of C in plants is paramount because it acts as an energy store and provides structure, representing most of the plant biomass, i.e., around 46% in leaves, 47% in stems, 45% in bark and woods, and 45% in roots. (Thomas and Martin, 2012; Ma et al., 2017). The concentrations of other elements like N and P play significant roles in plant nutrition and metabolic processes and act synergistically with C (Taiz et al. 2014). For example, N is essential for protein synthesis and chlorophyll formation, directly affecting photosynthesis and carbon fixation, while P regulates energy transfer via ATP, impacting carbon assimilation and growth (Hawkesford et al., 2012). Further, considering that the concentrations of elementomes differ across species and populations in response to environmental gradients, forest ecosystems distributed over climatic gradients are expected to vary in both their species composition and elementomes (Sardans et al., 2021; Vallicrosa et al., 2022).

Most studies analyzing ecosystem productivity found significant correlations with leaf elementomes (Fernández-Martínez et al., 2020; Šímová et al., 2019; Yan et al., 2023). However, aboveground or whole organism (including roots) elementomes should be more strongly correlated with forest functioning (e.g., forest production in biomass) since they encompass information about several functional traits other than those related to leaves (Schreeg et al., 2014; Xing et al., 2022; Zhang et al., 2018a). For example, positive relationships between N and P concentrations in different plant organs (e.g., roots, stems, branches, and leaves) are essential for tree growth and productivity (Ding et al., 2022). Thus, to consider the concentrations of aboveground elementomes, one should calculate them by weighing the elementomes of different organs by their relative biomass (Fernández-Martínez, 2022). However, to date, no study has assessed or compared the predictive performance of leaf versus whole or aboveground organism elementomes in predicting forest functioning.

Considering elements (concentrations and stocks) of the entire aboveground biomass and leaves only may contribute to enhancing the understanding of ecosystem processes (Luo et al., 2020; Rocha et al., 2011). Forest biomass production (i.e., the overall total amount of biomass accumulated over an area in a given period) is influenced by the concentration of elements the plants store (Dar and Parthasarathy, 2022; Ullah et al., 2024). Fine roots, for example, influence tree nutrient stocks since they regulate processes like water absorption and nutrient uptake from the soil (Likulunga. et al., 2022; Zhao et al., 2022). Further, tree elemental concentrations (e.g., from aboveground organs) significantly impact ecosystem productivity (Bitomský et al., 2023; Elser et al., 2010). Therefore, elemental concentrations and stocks also contribute to forest biomass productivity—a unit of biomass (e.g., per area and year) produced per unit of standing biomass that reflects ecosystem efficiency (Margalef, 1998; Lartigue and Cebrian, 2012).

Forest biomass productivity is also affected by the variation of elementomes in different stand ages, e.g., limited N and P content in older stands (Zhang et al., 2018a; Zhang et al., 2022). Different stand ages also shape the tree element stocks (i.e., elements stored within the biomass) in tree organs (Hoover and Smith, 2023; Rodríguez-Soalleiro et al., 2018). The variability of plant nutrient stocks, particularly C, N, and P, determines how trees allocate resources between roots and aboveground organs, ultimately impacting their biomass growth (Yan et al. 2016; Li et al. 2024). Therefore, assessing the effects of the tree nutrient stocks on forest biomass contributes to a better understanding of their adaptation to varying nutrient and environmental conditions (Peng et al., 2020). Nevertheless, the predictive performance of elementomes compared to element stocks in explaining forest functioning remains scarcely understood. Furthermore, it remains unexplored whether elementomes and element stocks predict forest functioning better than environmental factors (e.g., climate) and stand age.

Finally, the optimal elemental set (OES) — the minimum set (number and combination) of elements — for achieving the best prediction of organism and ecosystem functioning in general remains elusive. Most studies investigating elementomes in forested ecosystems only focused on C, N, P, and K (Sardans et al., 2017; Schreeg et al., 2014; Vallicrosa et al., 2022; Xing et al., 2022; Zhang et al., 2018b), while fewer ones have also included other important elements for the functioning of organisms and forest ecosystems, like Ca, S, and Mg (Sardans et al., 2016; Sardans et al., 2021, 2015; Bai et al., 2019; Huang et al., 2019). Acquiring knowledge on forest OES can improve predictions of forest functioning by increasing our mechanistic knowledge of how organisms and forest ecosystems work.

In this study, we used a database including forest elemental composition and biomass growth in the northeast of the Iberian Peninsula. This region is a suitable model for investigating topics related to OES, as it is composed of a notable environmental gradient (e.g., wide variations in climate and altitude) that influences the formation of distinct forest types (Sardans and Peñuelas, 2014). Variations in climate, soil nutrients, and species composition lead to differences in plant stoichiometry (e.g., balance in the C, N, and P) across distinct forest types, thus affecting their growth rates and biomass accumulation (Sardans and Peñuelas, 2014; Shi et al., 2016). Therefore, environmental gradients, such as the cited study region, allow for more robust assessments of general trends in the influence of OES on forest biomass growth. We aimed to answer four questions: Q1-Are the aboveground elements (elementomes and stocks) better predictors of forest functioning (biomass production and productivity) than only leaf elements? Q2-Do element stocks better explain forest functioning than elementomes? Q3-Do element stocks and elementomes (leaf and aboveground) explain better forest functioning than environmental factors and stand age? Q4-What is the OES that best predicts forest functioning? Related to these questions, we established three central hypotheses.: H1: Aboveground elements (elementomes and stocks) are better predictors of forest functioning (biomass production and productivity) than only leaf elements (Q1); H2: Element stocks better explain functioning than elementomes, as the former incorporates the effect of growth, while also encompasses effects of factors such as age and hidden limitations (e.g., carbon saturation, nutrient limitation), in forest functioning (Q2, Q3); H3: OES effects in forest biomass production and productivity models are greater in models using whole organisms than leaf elementomes (Q4). Answering the questions above can contribute significantly to enhancing the knowledge about the role of plant elementomes

in forest growth while providing practical insights for researchers and managers on which type of elemental data (e.g.,
aboveground elements or only leaves' elements) to collect and assess.

**2 Material and Methods**

**2.1 Study Area**

This study was conducted across the northeast of the Iberian Peninsula (ca. 31,900 km$^2$), bounded in the north by the
Pyrenees and in the east by the Mediterranean Sea. We chose this region due to its heterogeneous climatic conditions associated
with large ranges in altitude (i.e., 0 to > 3000 m) and distance from the sea, which together result in wide variations in mean
annual temperature (from 1 °C to 28 °C) and precipitation (annual mean from 350 to >1500 mm) (Martín Vide et al., 2008).
Further, the forests in this region exhibit a diverse range of soil types, predominating cambisols, fluvisols, regosols, and
leptosols (Soil Atlas of Europe, 2006; ICGC, 2019), with variations in organic matter and moisture content depending on the
specific forest area (Selkimäki et al., 2011). The Mediterranean climate is mostly characterized by mild winters, dry and warm
summers, and a high degree of interannual variability in precipitation. Such an array of environmental conditions in the study
region displays significant roles in variation in elemental allocation (e.g., N, P, K), thus influencing the nutrient stocks across
forest types (Sardans and Peñuelas, 2014). These pronounced climatic and soil gradients allow for the establishment of three
predominant forest types: Mediterranean evergreen angiosperm forests (dominated by *Quercus ilex* trees), Mediterranean
gymnosperms (stands of *Pinus halepensis*, *Pinus nigra*, *Pinus pinea*, *Pinus sylvestris*, *Pinus uncinata*, and often with *Quercus*
*petraea* and *Q. ilex* among them), and wet temperate deciduous angiosperms (with *Fagus sylvatica*, *Quercus faginea*, *Quercus*
*robur*, *Q. petraea*, *Abies alba*, and *P. sylvestris* dominating at altitudes from 800 to 1500 m and *P. uncinata* from 1600 to 2400
m) (García et al., 2004; Bolòs i Capdevila, 1991).

**2.2 Forest Inventory and Elemental Data**

We used the Ecological and Forest Inventory of Catalonia (IEFC) database, originally sampled in the period 1989-
1996 (Gracia et al., 2004) (http://www.creaf.uab.es/iefc). This database includes tree diameters, basal area, biomass, and annual
forest production of leaves, branches, barks, and stems, as well as the corresponding elemental composition of these organs.
The forest sites from which we compiled the data represent sampling plots (10 m radius) distributed throughout Catalonia. The
sampling was conducted at a density of one plot per square kilometer (sq km) of natural or managed forest (Gracia et al., 2004).
For plots having more than five tree species, only the five most abundant ones (DBH > 5 cm) were recorded, and a tree core
sample was used to calculate the stand age and annual tree growth over the last five years (Vilà et al., 2003). The estimation
of branch and leaf biomass was based on normalized dimensional analysis (Duvigneaud, 1971; Whittaker and Woodwell,
1969). The concentrations of the elements, i.e., elementomes (N, C, P, K, S, Mg, and Ca), of the individuals of each species

were measured for samples of the entire set of aboveground organs (i.e., wood, bark, branches, and leaves) by drying and grinding them to obtain homogeneous samples (Vayreda et al., 2016). Then, from an anhydrous subsample (oven-dried at 75 ºC) and of known weight, the concentration of nutrients was determined. The concentrations of C and N were determined by gas combustion chromatography in a C.E. elemental analyzer INSTRUMENTS (Wigan, UK). The concentrations of P, S, Mg, Ca, and K were determined by Inductively Coupled Plasma (ICP) in a Jobin Yvon JI-38 spectrophotometer (Edison, USES) (Vayreda et al., 2016). A complete description of the methods employed in this forest inventory (e.g., sampling procedures, allometric equations, data processing, etc.) can be found in Gracia et al. (2004).

From the IEFC dataset, we extracted the data regarding forest stand ages, biomass of tree individual organs, forest biomass production, and concentration of N, C, P, K, S, Mg, and Ca available for 2227 tree individuals (with a diameter at breast height (DBH) > 5 cm) from 48 species located in 2000 plots. The stand age is expressed in years and was obtained from the growth rings of tree wood cores in each plot (Gracia et al., 2004). In each plot, a core was taken from a tree that represented the center of the size class (diametric class), which was defined from each 5 cm increment DBH (e.g., 5–10 cm; 15-20 cm; 20–25 cm, etc.). Finally, it was calculated as the weighted average of the stand age based on the number of trees per DBH class. The elementomes of the trees were obtained for aboveground organs: leaves, branches, barks, and stems (data for roots are missing in the inventory). To access the procedures, parameters, and allometric equations used to calculate the biomass of each organ, please see the methodological details of the IEFC described in Gracia et al. (2004). In our analyses, we used forest biomass production calculated considering the following equation:

$P = (Bt^2 - Bt^1)/5,$

where $Bt^2$ is the current biomass (t ha$^{-1}$: tons per hectare) per area and $Bt^1$ is the biomass 5 years before (Vayreda et al., 2005; Vilà et al., 2003). Thus, forest production responds to the net increase in biomass in the ecosystem per year (t ha$^{-1}$ y$^{-1}$). Further, to obtain forest productivity (production per unit of standing biomass, y$^{-1}$), we summed the biomass of tree organs (leaves, branches, bark, and stem wood) to get the whole aboveground tree biomass. Then, we divided forest production by the aboveground tree biomass. Therefore, we emphasize that in our study, forest biomass production and productivity were measured considering only above-ground tree organs.

For our analyses (see section Statistical Analyses), we used values of concentration (g/100 g) and stocks of N, C, P, K, S, Mg, and Ca for only leaves and the entire set of aboveground organs. The aboveground elementome was calculated as the weighted average of the elemental concentration (g 100 g$^{-1}$) of the different plant organs. The stocks (t ha$^{-1}$) of the elements per organ were calculated as the biomass of the organ multiplied by the concentration of the element. Finally, we summed the stocks of each element from the different organs to obtain the aboveground stock.

**2.3 Climatic Data**

For each forest plot, we acquired data on the 19 bioclimatic variables provided by the WorldClim database version 2 at a very high spatial resolution (approximately 1 km$^2$) (Fick and Hijmans, 2017). From the 19 variables, we selected only the ones with coefficients of correlation < 0.70 (Dormann et al., 2013) to avoid biasing the statistical models (see the section Statistical Analysis) due to multicollinearity. Our final set of climatic variables was composed of temperature seasonality, mean temperature of the wettest quarter (three months), precipitation of the wettest month, precipitation of the driest quarter, precipitation of the warmest quarter, and precipitation of the coldest quarter.

**2.4 Statistical Analysis**

To test our hypothesis on the highest performance of aboveground elementomes and element stocks for predicting forest functioning (biomass production and productivity) compared to leaves or to environmental variables (climate) and stand age, we first constructed gaussian generalized additive mixed models (GAMM) using the R package "mgcv" (Wood, 2017). For predicting forest biomass production, we used five different models characterized by the following sets of predictors: i) aboveground elementomes; ii) aboveground element stocks; iii–iv) the same as items i and ii but for the leaves; and v) the environment (climate) and stand age. To predict forest productivity, we used three different models with the following sets of predictors: i) elementomes of the leaves; ii) aboveground elementomes ; and iii) the environment and stand age. The predictors representing elementomes and element stocks were N, C, P, K, S, Mg, Ca, and the interactions C×P, C×N, and N×P. For forest productivity, stocks were not included as predictors to avoid statistical redundancy since the productivity calculation involves the sum of organ biomass and stocks also use organ biomass (details in the Forest Inventory and Elemental Data section).

To adequately fit the GAMMs and eliminate spatial autocorrelation effects on the residuals, we included the coordinates (longitude and latitude) of the forest plots as fixed smoothed terms with Duchon splines (Duchon, 1977; Wood, 2003), while also adding species as random effects. This approach guaranteed that the degrees of freedom of the splines (Edf) were correctly fitted according to the required number of knots (k) for the GAMMs to reach residual independence. To verify whether potential spatial effects were sufficiently eliminated, the residuals extracted from the GAMMs were modeled in spatial variograms using the function "fit.variogram" of the R package "gstat" (Pebesma, 2004). We found no significant remaining spatial effect on the residuals of the models. Further, to achieve the normality of the residuals, we transformed the target forest production into its natural logarithm in all models. For the proper fit and convergence of the models regarding forest biomass productivity, we normalized (mean divided by the standard deviation) all elementomes using the built-in "scale" R function.

To find the OES of the elementome for predicting forest production and productivity and to discern whether leaf or aboveground elementomes work better for this purpose, we performed a model selection procedure based on the Akaike information criterion (AIC) (Burnham and Anderson, 2002). Such procedure consisted of including the global GAMMs (with the same eight models above described: five for production and three for productivity) in the function "dredge" of the "MuMIn" package (Bartoń, 2023) in the R programming environment version 4.3.3 (R Development Team Core, 2024). The use of the minimum AIC selection procedure allowed us to extract the best combinations (subsets) of predictors from our global models

to predict forest functioning. We applied the same selection procedure to models with the environment and age as predictors. In all selections, we considered the subsets with the lowest AIC values as the best models.

We also considered all subsets of selected models with delta (ΔAIC) < 4 as equally robust and statistically reliable, thus allowing us to retain relevant and valuable information beyond single-best models (Burnham et al., 2011). From these subsets (ΔAIC < 4), we extracted information on the performance of the models (R-squared) and the number of variables they selected. Then, we assessed the predictive performance (R-squared: R2) by accessing the models' outputs in two ways: by the subset models according to the number of selected predictors and by the overall performance only of the single best models. This two-way performance ranking allowed us to compare the performance of only the single best models (lowest AICs) with sets of models equally reliable (ΔAIC < 4).

Finally, to obtain a reliable overview of which were the most important variables (e.g., elements concentration and stocks) for explaining forest functioning, we performed model averaging for models with ΔAIC < 4 using the function "model.avg" of the "MuMIn" package (Bartoń, 2023) in R 4.3.3. We used the argument "beta=TRUE" to standardize the coefficients, allowing for a comparison of the relative importance of each predictor variable in the average models. Model averaging computes an average model output from the estimates of a set of models and weights their relative importance by their AIC (Burnham and Anderson, 2002). Therefore, this approach allowed us to obtain information on the importance of predictor variables extracted from the best model subsets (i.e., ΔAIC < 4).

The complete routine with the codes used to execute the models described and presented in this study can be accessed in Diniz (2024).

**3 Results**

By assessing the predictive performance of the best single models (lowest AIC; Table A1, Appendix A), we answered the questions regarding the performance of the aboveground (elementomes and stocks) *vs.* leaves and of the elementomes *vs.* stocks for explaining forest functioning. Our results indicated that leaves (rather than aboveground) and stocks (rather than elementomes) are the best predictors of forest biomass production and productivity. We found that the best model of forest biomass production using leaf element stocks as predictors explained 58% of the variance and had nine variables: C, Ca, K, Mg, N, P, C×N, C×P, and N×P (Fig. 1a). The second-best model explained 28% of the variance of forest biomass production (Fig. 1a) and had three aboveground element stocks as predictors (C, N, and C×N). Regarding the best models of forest production, including elementomes as predictors, we found that leaf elementomes also explained more variance (22%) than aboveground elementomes (13%) Fig. 1a). The best leaf elementome model included six variables (C, Ca, N, P, C×P, and N×P), and the best aboveground elementome model included only one (Ca). Forest biomass productivity was best predicted by the model with climate and stand age as predictors (Fig. 1c, d). Secondarily, between leaf elementomes (Ca, K, and N) and aboveground elementomes (K), the first ones were the best predictors of forest biomass productivity (Fig. 1c; 28% of variance explained).

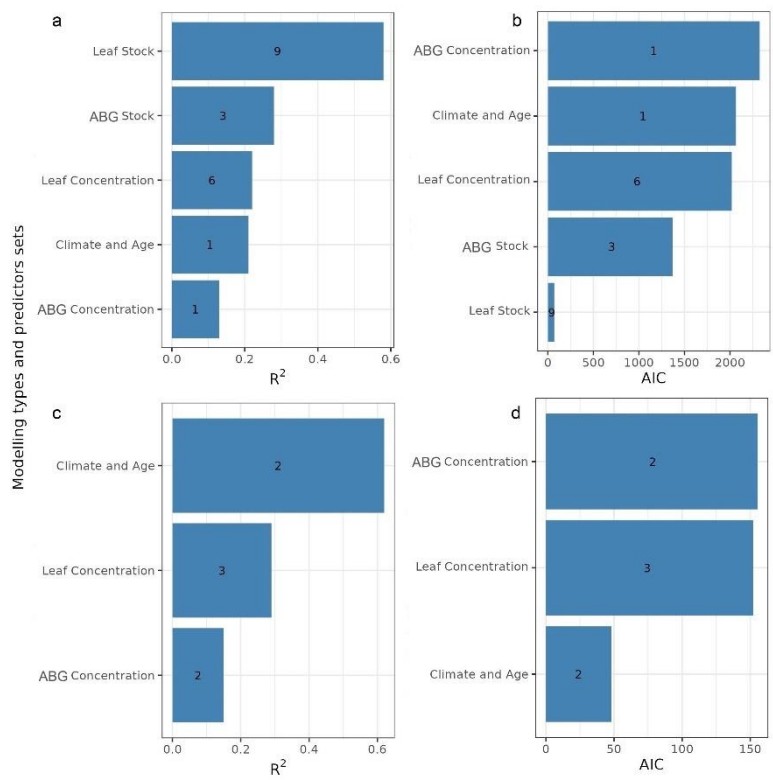

234

**Figure 1: R² and AIC of the best models for explaining forest biomass production (a, b) and productivity (c, d), considering as predictors the stocks and the concentration of elements only for the leaves and for the entire set ofaboveground plant organs, and climate and forest age. Numbers within the bars show the number of variables selected. ABG concentration = aboveground elementomes.**

Our subsets of models, equally robust (ΔAIC< 4), showed that the optimal elemental set (OES) for predicting forest biomass production from leaf element stocks (Fig. 2a) was nine variables (C, Ca, K, Mg, N, P, C×N, C×P, and N×P). This model subset explained an average of 58% of the variance in forest biomass production. The subset of models using aboveground element stocks exhibited the second-best predictive performance for forest biomass production ($R^2 = 0.29$; Fig. A1, Appendix A). Differently, the subset of models using climatic variables and aboveground elementomes as predictors displayed the lowest prediction of forest biomass production (Fig. A1). The variance of forest productivity was moderately explained (28%) by models selecting three variables (Ca, K, and N) of leaf elementomes (Fig. 1c, d) and poorly explained (15%) by models with aboveground elementomes (Fig. A2, Appendix A). Forest productivity was best explained ($R^2 = 0.68$) with the subset of models that included two variables (temperature seasonality and stand age) (Fig. A2).

We also found that climate and stand age (Fig. A1, Appendix A) explained 21% of the variance in forest biomass production, while leaf element stocks explained 58% (Fig. 1a and 2a). On the other hand, the best subset of models that had

forest age and temperature seasonality as predictors displayed the best performance and explained 62% of the variance in forest
biomass productivity (Fig. A2, Appendix A).

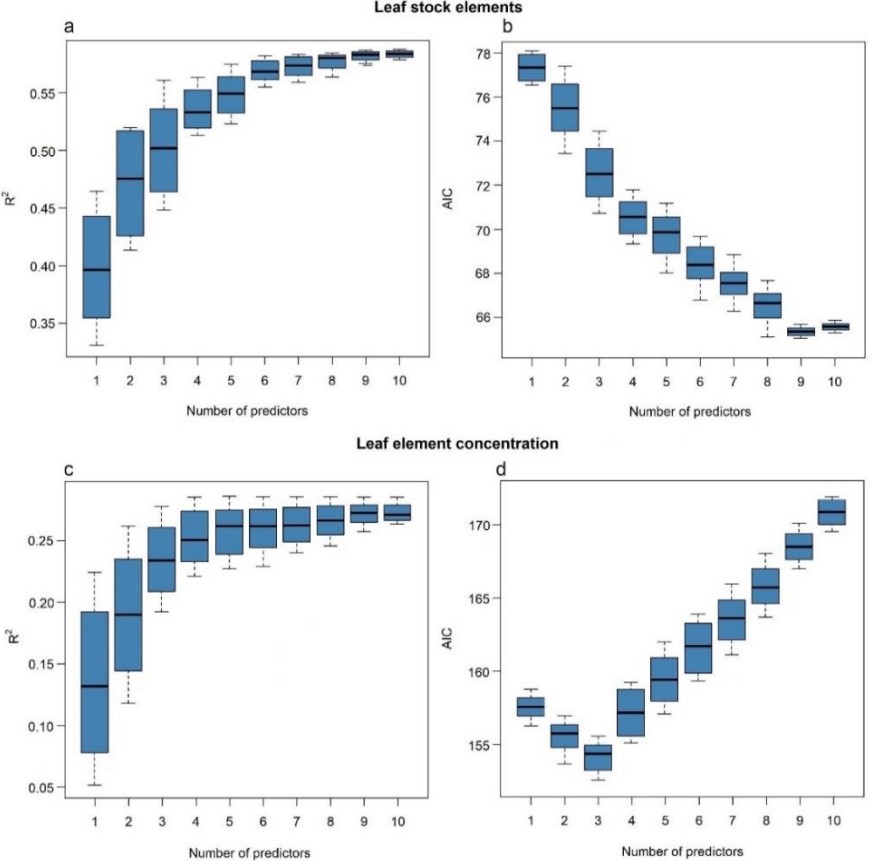


**Figure 2: Forest biomass production (a, b) and productivity (c, d) predicted by leaf element stocks (a, b) and leaf**
**element concentration (c, d). Results demonstrated by the performance (AIC and $R^2$) of the most robust subsets of**
**models ($\Delta$AIC < 4).**

258         The information contained in Figures 3, 4, and A3 outlines the importance of individual elements (concentrations and

stocks) in contributing to the performance of models in predicting forest functioning. The average models are based on different
subsets of variables (i.e., leaves vs. aboveground elementomes and stocks, and elementomes vs. stocks; Table A2, Appendix
A) and demonstrated that P, Ca, and N — from both models based exclusively on leaf element stock and models only with
leaf elementomes — are the most important predictors for explaining spatial variability in forest production (Fig. 3 a, c; Fig.
A3, Appendix A). Conversely, the aboveground elementomes and element stocks of the P exerted a low and non-significant
influence on forest biomass production (Fig. 3 b, d). N stocks (leaves and aboveground) and N leaf concentration were
positively correlated to forest biomass production (Figures 3 a, b, and c, respectively; Fig. S3). On the other hand, in leaves,
the interactions N×P (Fig. 3a) and C×P (Fig. 3c) and the concentration of C (Fig. 3 c) exerted a significant and negative effect
on biomass production. The negative interaction of N×P indicated that the higher the value of P, the lower the effect of N on
biomass production. Similarly, the negative interaction of C×P implied that higher values of P reduce the effect of C on biomass
production. The average models using leaf and aboveground predictors were unable to predict forest biomass productivity
(Fig. 4).

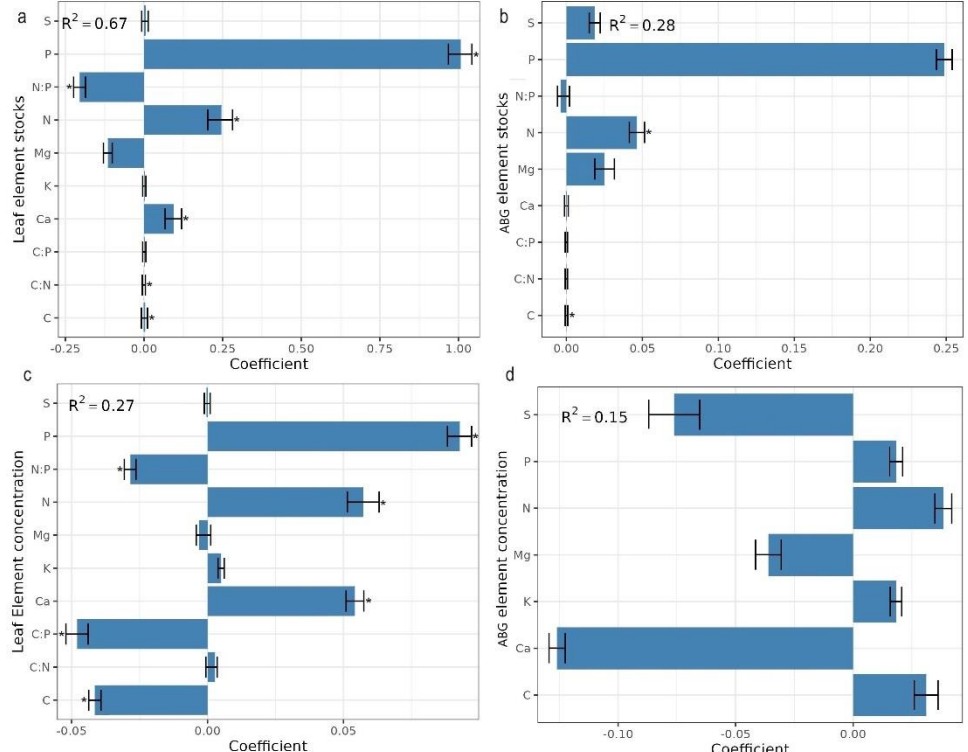


**Figure 3: Standardized coefficients from the model averaging (ΔAIC< 4) for the prediction and explanation of forest**
**biomass production, considering as predictors the stocks (a, b) and the concentration (c, d) of elements only for the**
**leaves (a, c) and for the entire set of aboveground plant organs (b, d). $R^2$ is the average of R-squared derived from all**
**models with ΔAIC < 4. ABG element concentration = Aboveground element concentration. * Indicates significant**
**coefficient.**

278       Climatic variables also displayed significant effects on forest biomass production. Temperature seasonality and
precipitation in the coldest quarter were negatively correlated with biomass production (Fig. 5a). Conversely, precipitation in
the driest quarter correlated positively with biomass production (Fig. 5a). However, forest biomass productivity was not
influenced by climate but decreased significantly with stand age (Fig. 5b).

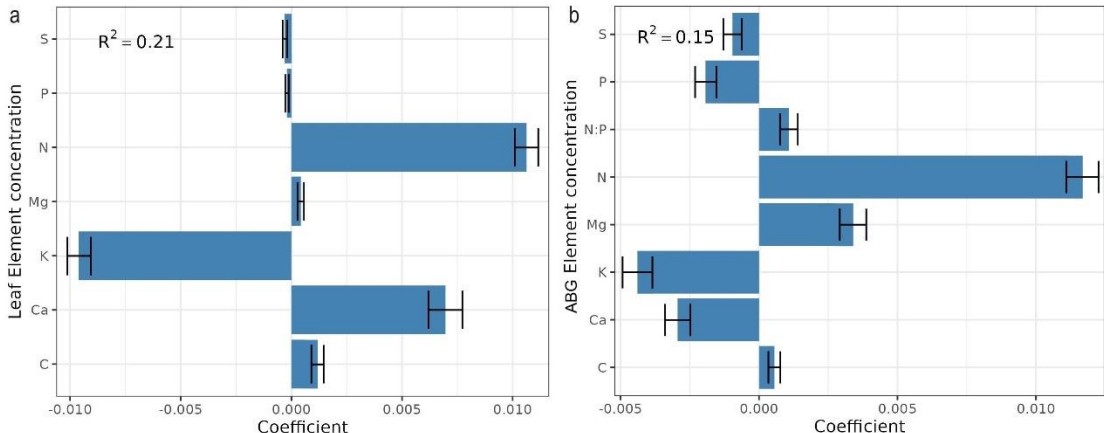


**Figure 4: Standardized coefficients from the model averaging (ΔAIC < 4) for the prediction of forest biomass productivity, considering as predictors the concentration of elements only for the leaves (a) and for the entire set ofaboveground plant organs (b). $R^2$ is the average of R squared derived from all models with ΔAIC < 4. ABG element concentration = Aboveground element concentration. * Indicates significant coefficient.**

287

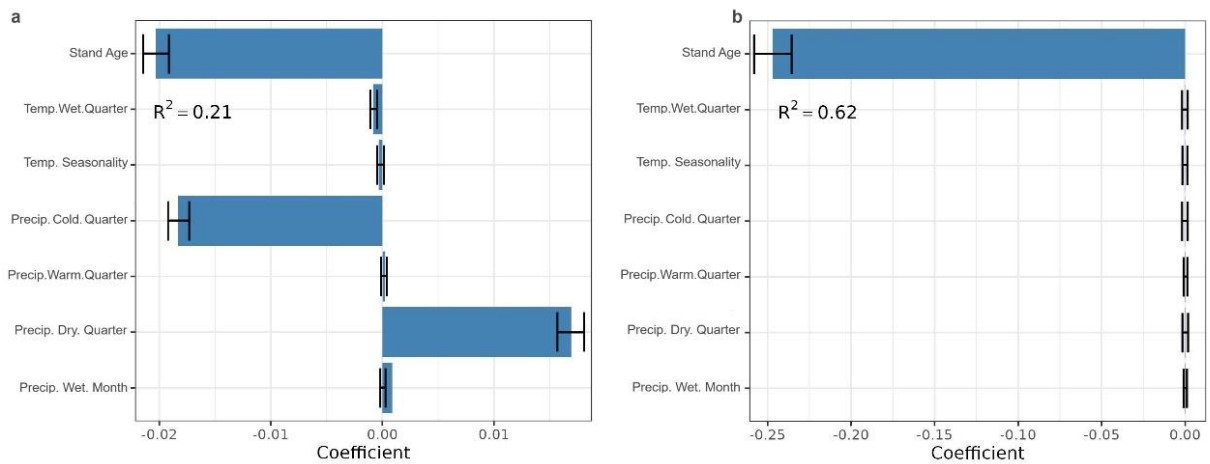

288

**Figure 5: Standardized coefficients from the model averaging (ΔAIC < 4) for the prediction of forest biomass production (a) and productivity (b), considering as predictors climate variables and stand age. Temp. Wet. Quarter: Mean temperature of the wettest quarter; Temp. Seasonality: Temperature Seasonality; Precip. Cold. Quarter: Precipitation of Coldest Quarter; Precip. Warm. Quarter: Precipitation of Warmest Quarter; Precip. Dry. Quarter: Precipitation of Driest Quarter; Precip. Wet. Month: Precipitation of Wettest Month. $R^2$ was averaged from all models with ΔAIC < 4. * Indicates significant coefficient.**

295

296

**4 Discussion**

We refuted the hypothesis that using aboveground elementomes and element stocks predicts forest biomass production better than leaf elementomes and element stocks alone. Models including nine leaf element stocks (C, Ca, K, Mg, N, P, C×N, C×P, and N×P) displayed the highest performance in predicting forest biomass production. On the other hand, stand age was the best predictor of forest biomass productivity. Altogether, these findings suggest that forest production can be best predicted by foliar element stocks and biomass productivity by stand age. Further, our average models indicate that changes in forest biomass production are mostly explained by concentrations and stocks of Ca, P, and N.

Our finding that leaf element stocks are the main predictors of forest biomass production was unexpected. Since the aboveground level considers different parts of the plant (e.g., stems, branches, bark) that require different nutrient concentrations to exert distinct functions (e.g., uptake, transport, storage), we could expect that using aboveground element concentrations and stocks aboveground would have higher predictive performance (Zhang et al., 2018; Delpiano et al., 2020; Sardans et al., 2023) than only using elements of leaves. However, even though the leaves do not encompass the whole functional space of a tree, they represent the essential photosynthetic part of a plant and the capability of rapid nutrient cycling and responsiveness to environmental conditions (Foster & Bhatti, 2020). For instance, N and P, the most important elements limiting plant growth, are more readily available in leaves for use in metabolic (e.g., growth) and ecosystem processes (e.g., biomass production) than in other organs (Liu et al., 2019; Roth-Nebelsick & Krause, 2023; Töpfer, 2021). Thus, the practical implication of our results for further studies is that foliar element stocks may hold sufficient information to derive robust predictions of forest functioning.

Foliar nutrient stocks are crucial for enhancing plant fitness by enhancing photosynthesis and thus biomass production (Gilliham et al., 2011; Taiz et al., 2014; Beechey-Gradwell et al., 2020). Sufficient reserves of macronutrients such as K, Ca, and Mg in specific leaf cell types are also vital for plant growth (Gilliham et al., 2011). The positive effect of the combination of stored elements on growth is indicated by our best model for biomass production, which had as predictors the foliar stocks of C, Ca, K, Mg, N, P, C×N, C×P, and N×P. Further, our average models also indicated the leaf stocks of Ca, P, and N as the most important predictors of forest biomass production.

The superior performance of leaf element stocks, compared to aboveground element stocks and concentrations, also might be due to suitable environmental conditions resulting in increased foliar biomass (Rodríguez-Soalleiro et al., 2018b; Urbina et al., 2011). In suitable climatic conditions (e.g., high precipitation), plant growth might be positively affected by high concentrations of foliar N and P (Kerkhoff et al., 2005; P. Reich and Oleksyn, 2004; Sardans and Peñuelas, 2014). We found a positive effect of precipitation in the driest quarters, N and P, on forest biomass production. Since, during summer, most of the territory addressed in this study coincides with high temperatures and marked water stress (Martín Vide et al., 2008), plants may invest in a strategy of retaining larger foliar nutrient reserves to cope with drought (Waring, 1987.; Gessler et al., 2017). Increased precipitation might enhance the foliar nutrients stored in drier periods, thus contributing positively to aboveground biomass production (Fernández-Martínez et al., 2017; Lie et al., 2018; Roa-Fuentes et al., 2012). In our study region, high

water availability (e.g., precipitation) correlates positively with mineralization, which enhances the nutrient availability to trees and contributes to increasing their biomass (Sardans et al., 2008).

The highest predictive performance was achieved by using foliar stocks including C, Ca, K, N, Mg, and P as predictors, which is congruent with the known high influence of the uptake and redistribution of these elements in forest biomass production (Bond, 2010; Whittaker et al., 1979). Such an optimal set of elements is influenced by the effects of climate and stand age on their uptake, redistribution, and storage (Woodwell et al., 1975; Augusto et al., 2008; Rodríguez-Soalleiro et al., 2018; Dynarski et al., 2023; Li et al., 2021). Thus, the driving role of climate in the optimal elemental set is expected to influence forest functioning ultimately. Indeed, we found that climate (precipitation in the driest quarter and temperature seasonality) correlated positively and significantly with biomass production. These findings suggest climate as the main factor that influenced the optimal combination of foliar stocks of C, Ca, K, Mg, N, P, C×N, C×P, and N×P in predicting biomass production (X. Wang et al., 2022; Yang et al., 2019; Q. Zhang et al., 2021).

Among the elements in the abovementioned optimal combination for predicting forest biomass production, N and P stand out. We found that higher leaf stocks of N and P were related to higher biomass production. Plant growth is highly influenced by the proportions of N and P, and particularly by the ratios N:P (Ågren, 2008; Gusewell, 2004; Sardans et al., 2011; Willby et al., 2021). The plant N:P ratio reflects the balance between uptake and loss of N and P (Gusewell, 2004). Our negative interaction with N×P indicates that the higher the leaf stocks of P, the lower the effect of N leaf stocks on biomass production. Such a higher importance of P compared to N for biomass production might be due to the typically higher foliar resorption of P than of N (Vergutz et al., 2012; Mulder et al., 2013).

The highest importance attributed to P for explaining forest biomass production is probably an outcome of its continuous storage in the forest biomass (Sardans and Peñuelas, 2015; Y. Wang et al., 2022). Thus, the observed prominent role of P might represent a long-term adaptive strategy of trees to store it in biomass and slow its loss from ecosystems (Sardans and Peñuelas, 2015). Sardans and Peñuelas (2015), using data from the Catalan Forest Inventory, found that trees with high woody biomass (branches plus stems) hold a higher P content than N and a higher P:N ratio with forest aging.

Aside from N and P, Ca also displayed a positive effect on forest biomass production and productivity, which is congruent with the importance of this element for photosynthesis, nutrient absorption, and plant growth (Hirschi, 2004; Ågren, 2008; Hochmal et al., 2015). However, the average models indicated that the concentration of elements (e.g., Ca and N in leaves and the entire set of aboveground organs) and climate were not significantly influential on biomass productivity. Rather, we observed a significant negative relationship between stand age and forest biomass productivity, probably explained by the increase of forest biomass and the decrease of forest nutrient availability with age (Fernández-Martínez et al., 2014; Goulden et al., 2011).

Lastly, the lower relevance of C in our average models may be partially due to its variations across distinct plant organs, e.g., the predominance of leaf and fine-root turnovers in C allocations (Yu et al., 2017). Besides, foliar nutrients, particularly P, significantly impact photosynthetic C uptake in forests, promoting variation in biomass production (Mercado et al., 2011). This leads to decreased biomass production in other organs, such as stems and barks (Jonsson et al., 2020; Ryan

et al., 1997; Schoonmaker et al., 2016; Yu et al., 2017). However, although plant biomass contains around 50% carbon, its production is not directly proportionate to C availability (He et al., 2020). Changes in N and P concentrations—important elements for regulating critical metabolic processes (e.g., protein synthesis, energy transfer)—may shift C allocation to maintenance and fine-root turnover, limiting structural biomass growth in stems and barks (Bruner et al., 2013; Likulunga et al., 2022). Consequently, other plant organs may allocate less C and reduce their biomass, ultimately limiting forest biomass productivity (Bruner et al., 2013; Neumann et al., 2020). Additionally, with growing P constraints under global change scenarios, C allocation patterns are projected to become more complex, directly reducing forest biomass production (Köhler et al., 2023).

**Caveats, limitations, and implications**

In this study, we bring new insights into the effects of the optimal elemental sets, compared to climate and stand age, on both forest biomass production and productivity. As practical implications for future research, our results suggest that using only data on leaf elements, especially stocks, allows us to achieve robust predictions of variations in forest biomass. Such information can contribute to decision-making by researchers and forest managers about the types of data (aboveground elements or only leaves' elements) they should prioritize collecting when assessing forest growth. Nevertheless, our presented results might be influenced by sampling limitations and analyses conducted only on aboveground organs (barks, branches, leaves, and stems). In the data used in this study, measurements of element concentrations in different above-ground organs of trees were obtained for various numbers of individuals per species. This difference in the number of individuals may have influenced, even if subtly, the results. Besides, the biomass of belowground organs (e.g., fine and coarse roots) may account for at least 22% of the total forest biomass (Ma et al., 2021) and display important roles in nutrient uptake and storage (Gao et al., 2021; Dybzinski et al., 2024). For some Mediterranean species, belowground organs may represent up to 50% of the forest biomass (Fernández-Martínez et al., 2014). Therefore, below-ground biomass and elementomes may help explain above-ground production and productivity. The importance of roots for element stocks is also underscored by the fact that around 24% of total plant carbon is stored belowground (Ma et al., 2021). Root biomass is also influenced by climatic factors such as temperature, thus leading us to expect that future changes driven by warmer and drier climates will affect the balance between aboveground and belowground biomass allocations and element stocks (Pornon et al., 2019; Ma et al., 2021). Alongside roots, soil nutrient stocks are also important contributors to forest biomass, since these stocks influence the construction of foliage and wood components (Zarzosa et al., 2021; De Vos et al., 2015; Augusto et al., 2017). Soil nutrient availability directly influences aboveground organs (e.g., leaves) nutrient stocks by driving nutrient uptake and allocation, which controls photosynthesis and biomass accumulation (Augusto et al., 2022; Wiesmeier et al., 2019). Thus, including element concentrations and stocks of roots and soil nutrients (concentrations and stocks) in statistical models may enhance the predictability of forest functioning. We suggest that future research includes belowground and soil elements in addition to

elements in aboveground biomass, to allow for the comparison between the predictive performance using whole-plant elements
(above and belowground) and only aboveground elements.

**5 Conclusions**

We found that elemental concentrations and stocks of leaves predict forest biomass production and productivity better
than those of the entire aboveground set of plant organs. Leaf stocks explained the highest amount of variance in forest biomass
production, thus suggesting that element stocks are better predictors than element concentrations. The optimal elemental set
for predicting forest biomass production can be achieved using leaf elemental stocks of C, Ca, K, Mg, N, P, C×N, C×P, and
N×P as predictors. Among these elements, N and P stocks and concentrations were the most positively correlated with biomass
production. Conversely, the concentration of elements and climate did not significantly influence forest biomass productivity,
which was mainly driven by stand age. Altogether, our results indicate that leaf element stocks are critical predictors of forest
biomass production.

**Code and Data Availability**

The data used in this study are maintained by the CREAF institute and are available upon request. Complete
information about the data and instructions for requesting its use can be accessed at the link: http://www.creaf.uab.es/iefc/.
Codes used to produce the models are provided by Diniz (2024).

**Author Contribution**

Écio Souza Diniz: Conceptualization, Methodology, Validation, Formal analysis, Investigation, Visualization,
Writing - original draft, Writing - review & editing. Eladio Rodríguez Penedo: Data Processing, Formal analysis, Writing –
review. Roger Grau-Andrés: Methodology, Validation, Writing - review. Jordi Vayreda: Data curation, Writing - review.
Marcos Fernández-Martínez: Methodology, Validation, Supervision, Visualization, Project administration, Writing – review,
Funding Acquisition.


**Competing Interests**

The authors declare that they have no conflict of interest.

**Acknowledgements**

This research was supported by the European Research Council project ERC-StG-2022-101076740 STOIKOS and the Spanish Research Agency (AEI) project ETRAITS (PID2022-141972NA-I00). M.F-M. was supported by a Ramón y Cajal fellowship (RYC2021-031511-I) funded by the Spanish Ministry of Science and Innovation, the NextGenerationEU program of the European Union, the Spanish plan of recovery, trans- formation and resilience, and the Spanish Research Agency. This paper is dedicated to those who conducted the Ecological and Forest Inventory of Catalonia (IEFC) displaying works in the field, office, and lab. The IEFC was financed by the "Departament d'Agricultura, Ramaderia i Pesca" and the "Departament de Medi Ambient de la Generalitat de Catalunya.

440

**Funding Source**

442

This research was supported by fundings provided by: European Research Council project ERC-StG-2022-101076740-STOIKOS, the Spanish Research Agency (AEI) project ETRAITS (PID2022-141972NA-I00), and Ramón y Cajal fellowship (RYC2021-031511-I).

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

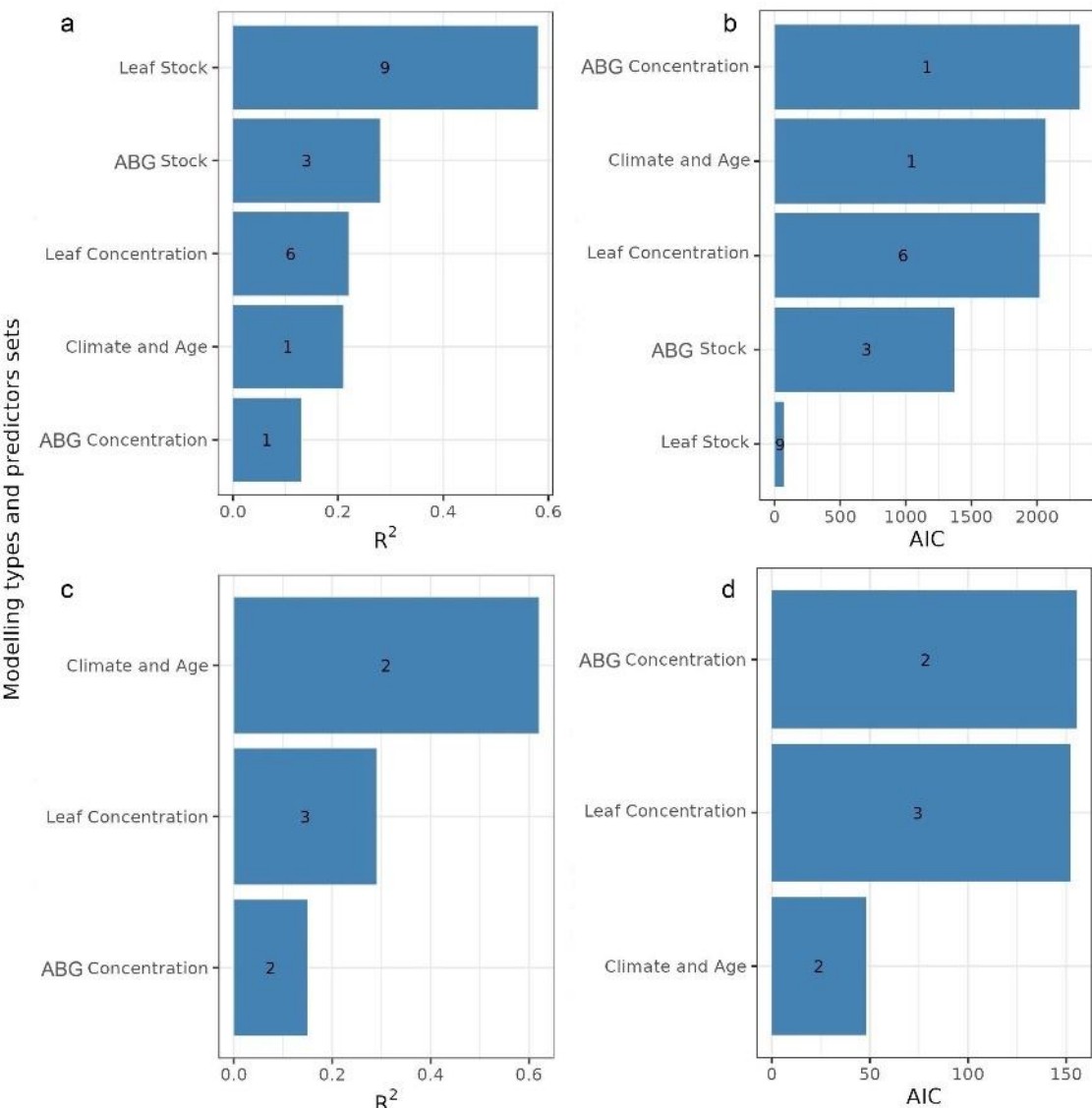

**Figure 1**



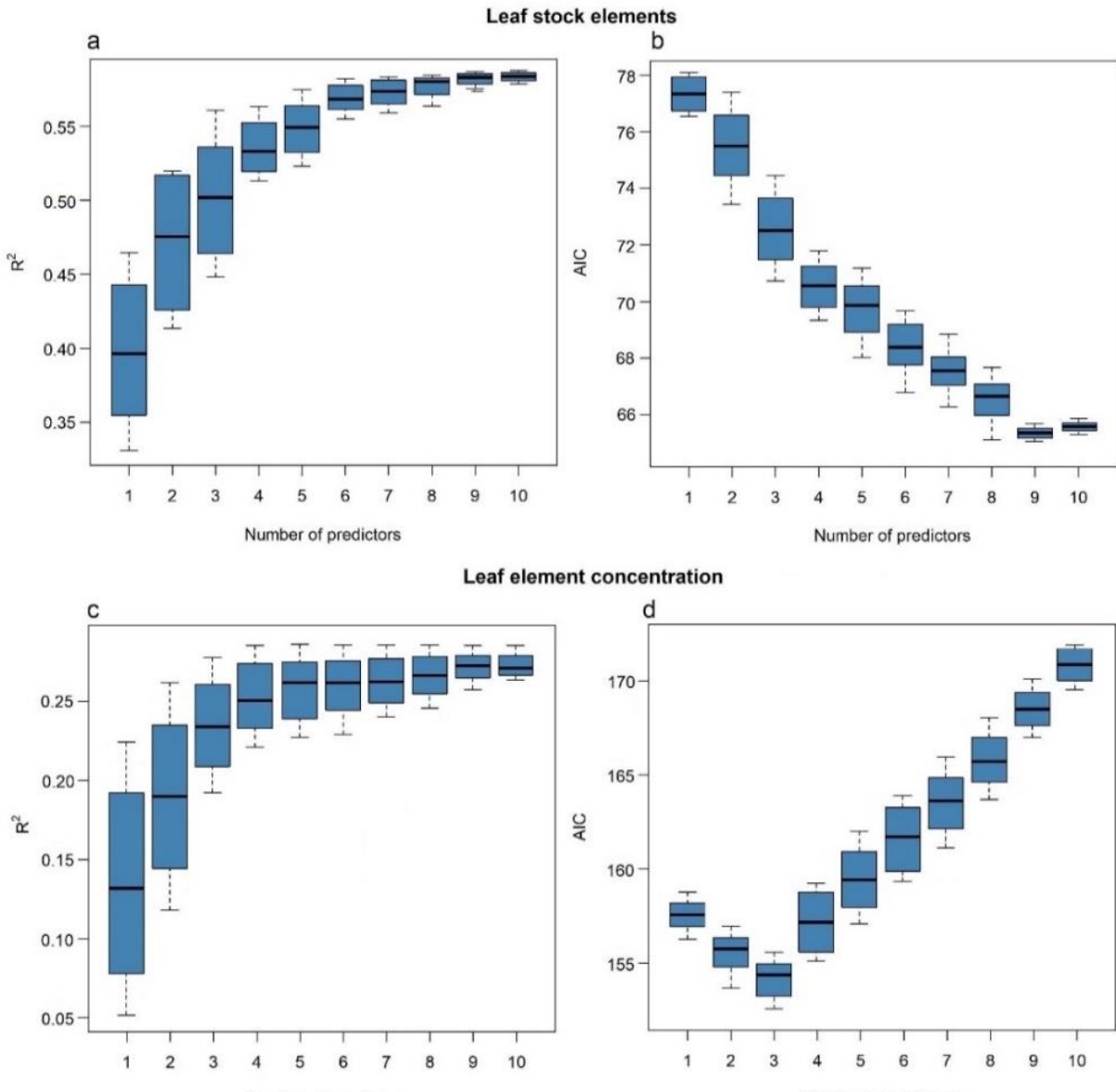


**Figure 2**








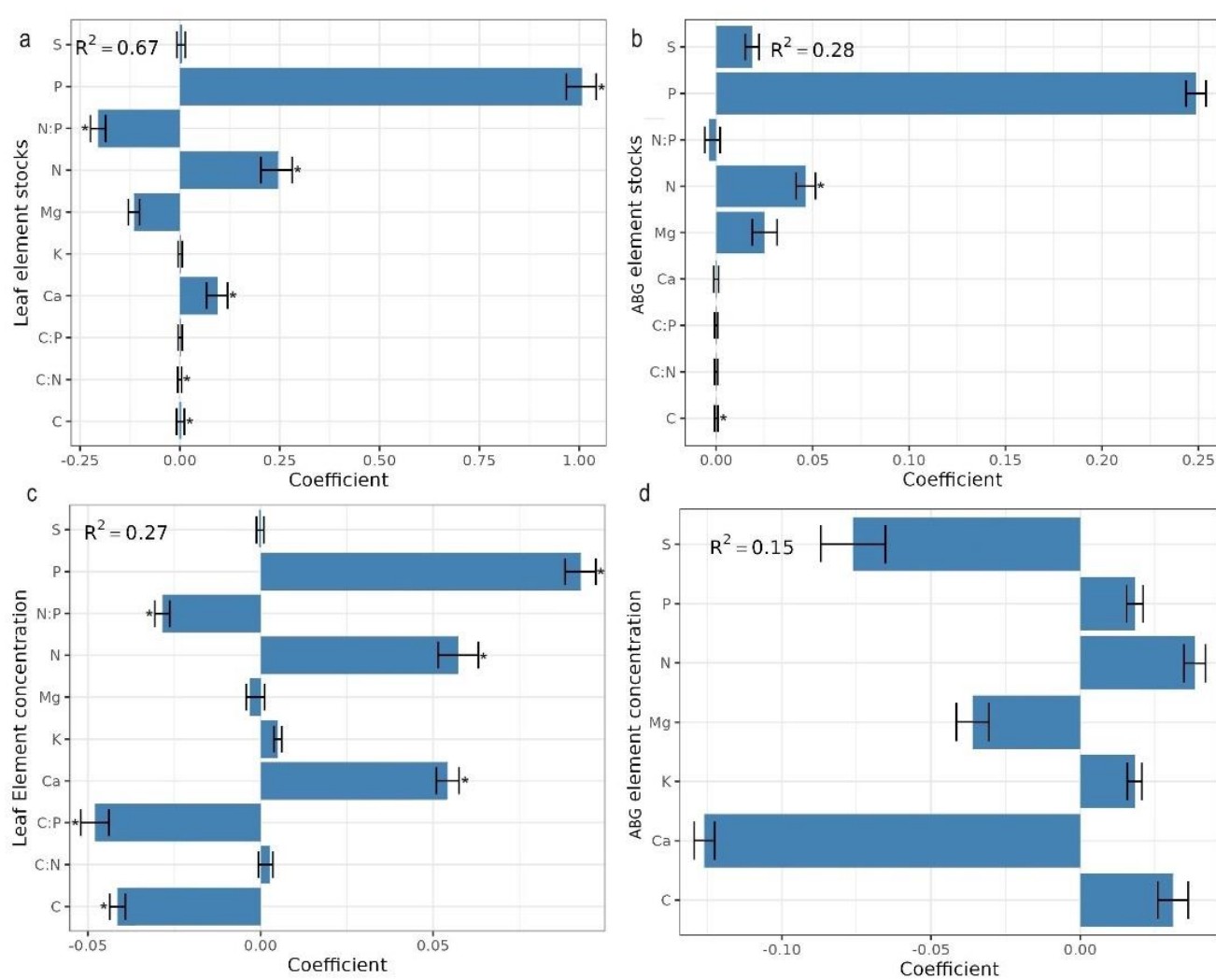


**Figure 3**










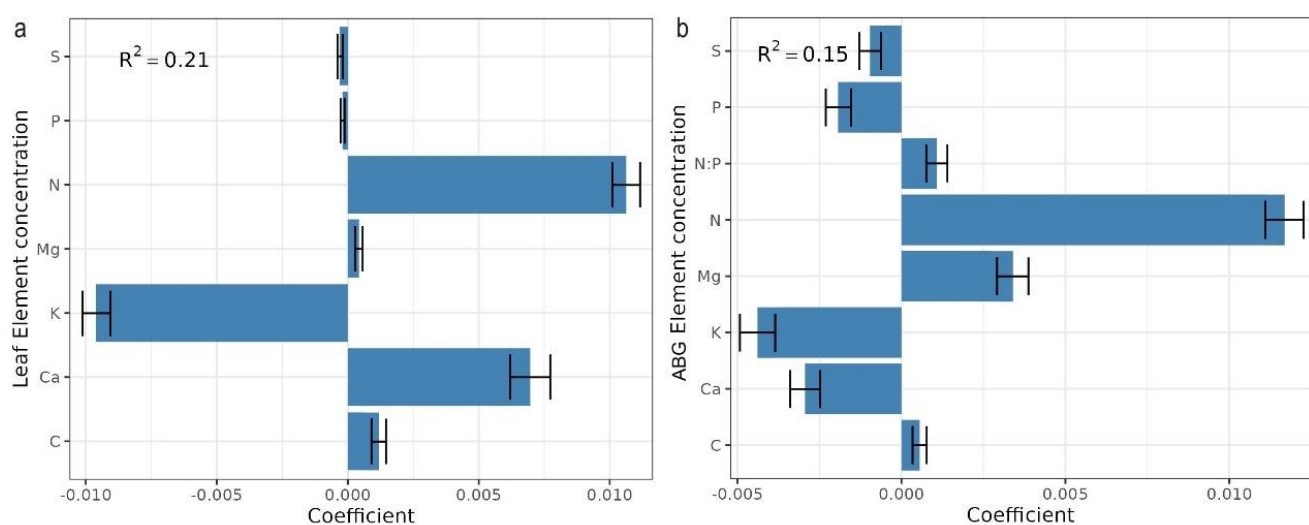


**Figure 4**








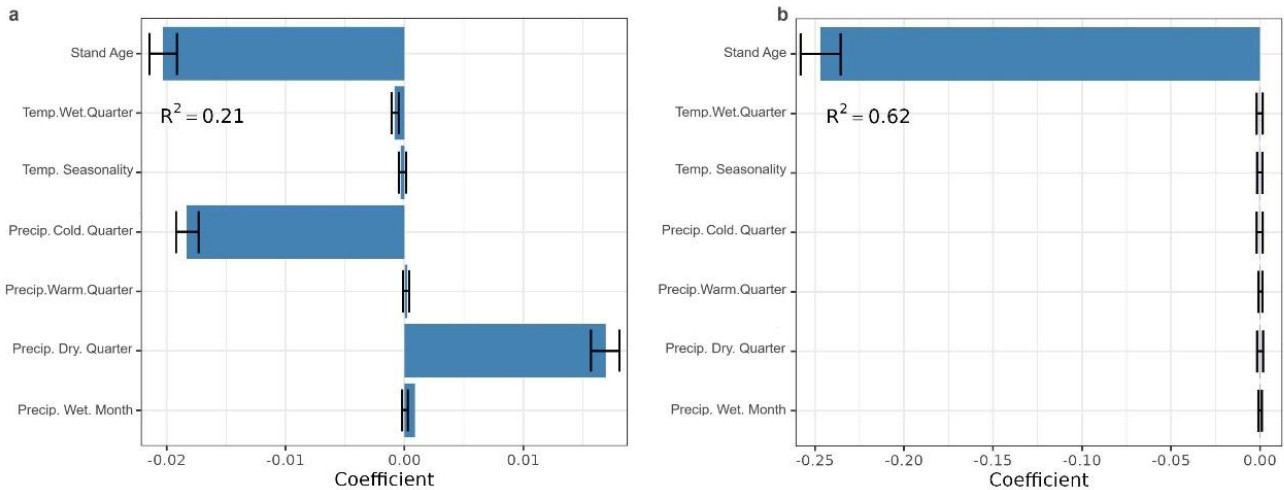

**Figure 5**

**Appendix A: Model Performance**

**Table A1: Performance of the best models (lowest AIC) showed in Figure 1 and the numbers (N) of predictors they selected for predicting forest production and productivity. Response = dependent variable. Leaf Conc. and ABG Conc. are leaf element concentration and aboveground plant element concentration, respectively. Clim. Age are climatic variables and stand age. Temp. Season = Temperature Seasonality; Temp. Wet. Quart. = Mean Temperature of Wettest Quarter; Prec. Dr. Quart. = Precipitation of Driest Quarter; Prec. Cold.Quart. = Precipitation of Coldest Quarter; Age = Stand age.**

| Response | Predictors | N | R2 | AIC | Selected variables |
|---|---|---|---|---|---|
| Production | Leaf Stock | 9 | 0.58 | 64.7 | C, Ca, K, Mg, N, P, C×N, C×P, and N×P |
| Production | ABG Stock | 3 | 0.28 | 1369.2 | C, N, and C×N |
| Production | Leaf Conc. | 6 | 0.22 | 2019.4 | C, Ca, N, P, C×P, and N×P |
| Production | ABG Conc. | 1 | 0.13 | 2326.2 | Ca |
| Production | Clim. Age | 1 | 0.21 | 2066.1 | Temp. Season., Temp. Wet. Quart., Prec. Dr. Quart., Prec. Cold.Quart. |
| Productivity | Leaf Conc. | 3 | 0.28 | 152.2 | Ca, K, and N |
| Productivity | ABG Conc. | 2 | 0.15 | 155.5 | K |
| Productivity | Clim. Age | 2 | 0.62 | 48.1 | Temp. Season., Age |

**Table A2: Total number (Total N) of models' subsets produced by the selection with "dredge" using different**
**predictors' set for predicting forest production and productivity. N (ΔAIC<4) is the number of models equally robust**
**under ΔAIC < 4 and used to calculate the average models. ABG Concentration and ABG Stock are aboveground**
**concentration and aboveground stock, respectively**

| Target | Predictors | Total N | N (ΔAIC<4) |
|---|---|---|---|
| Production | Leaf Stock | 575 | 10 |
| Production | ABG Stock | 575 | 10 |
| Production | Leaf Concentration | 852 | 10 |
| Production | ABG Concentration | 852 | 8 |
| Production | Climate and Age | 511 | 7 |
| Productivity | Leaf Concentration | 850 | 7 |
| Productivity | ABG Concentration | 850 | 8 |
| Productivity | Climate and Age | 511 | 7 |


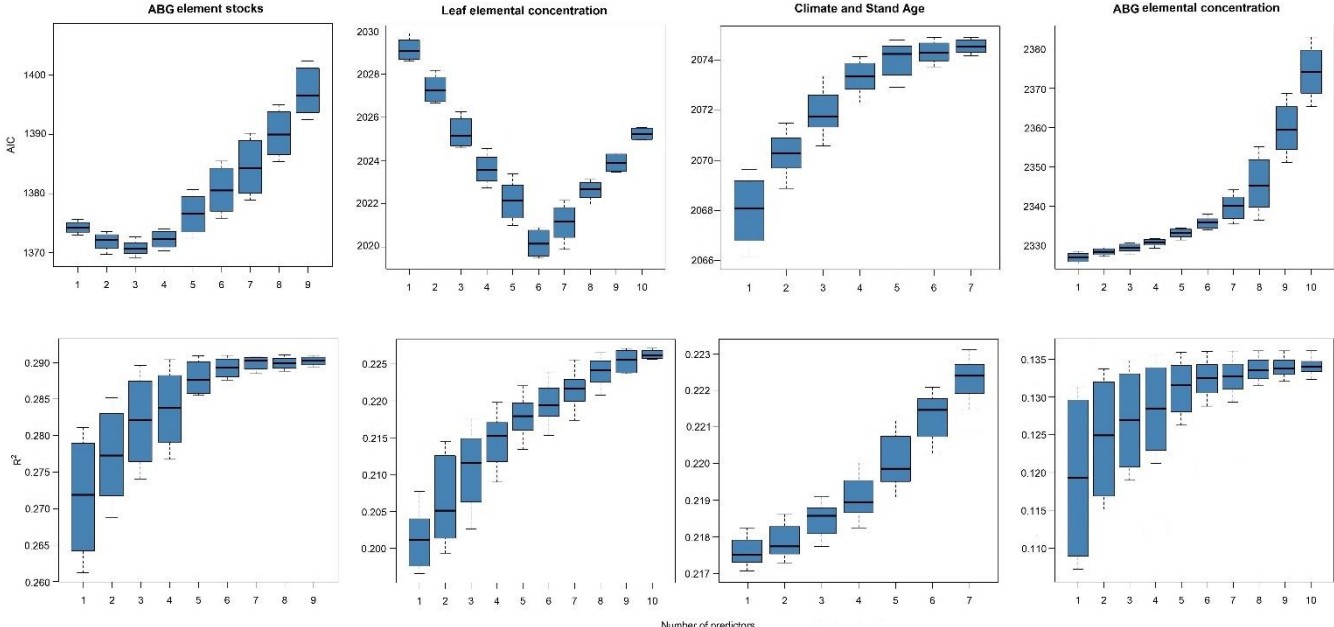

**Figure A1: Performance (AIC and R2) of the most robust models (ΔAIC < 4) in predicting forest production according to the number of selected predictors. The models' performance demonstrated by their AIC and R2: Plant stocks (a, e); Leaf elemental concentration (b, f); climate and stand age (c, g); Aboveground (ABG) elemental concentration (d, h).**


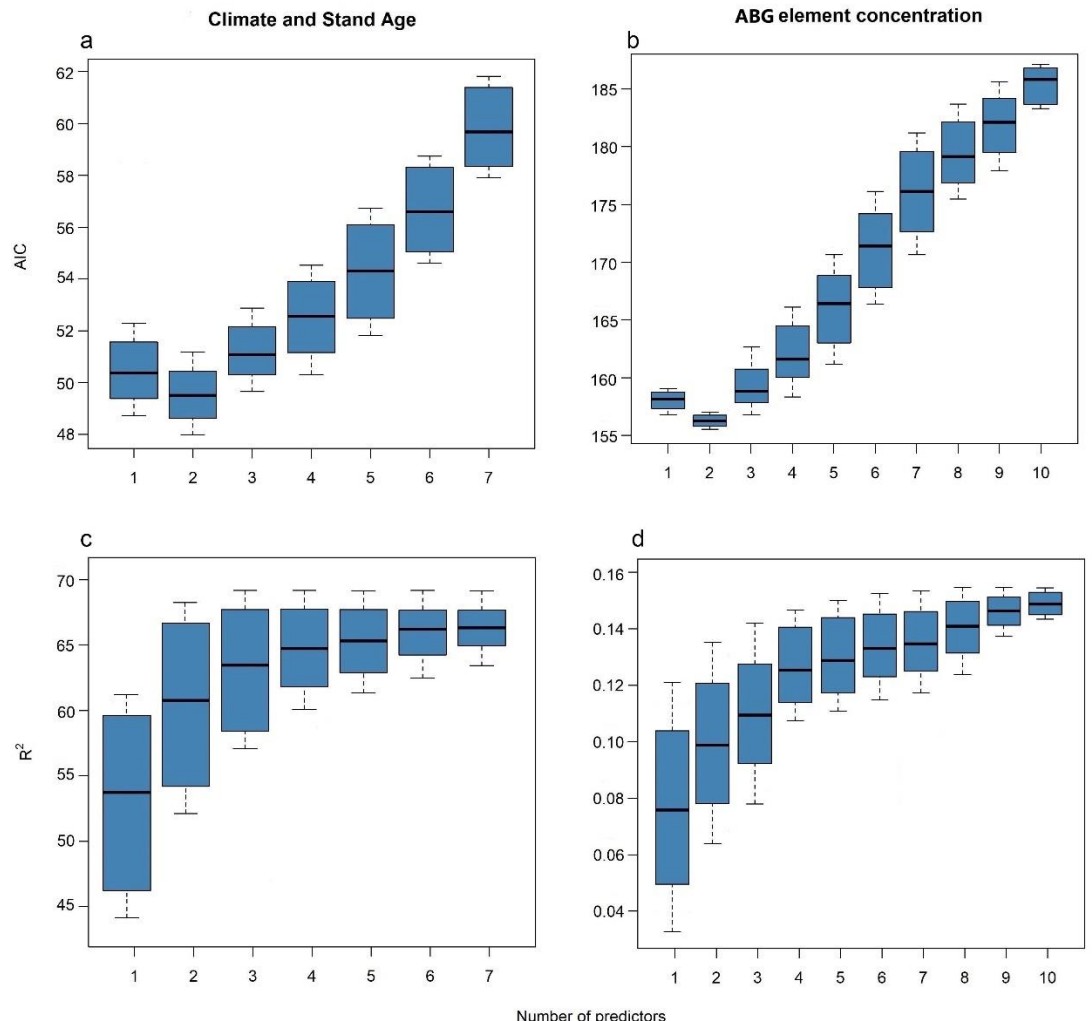


**Figure A2: Performance of the most robust models (ΔAIC < 4) in predicting forest productivity according to the number of selected predictors. The models' performance demonstrated by their AIC and R-squared: climate and stand age (a, c); Aboveground (ABG) elemental concentration (b, d).**











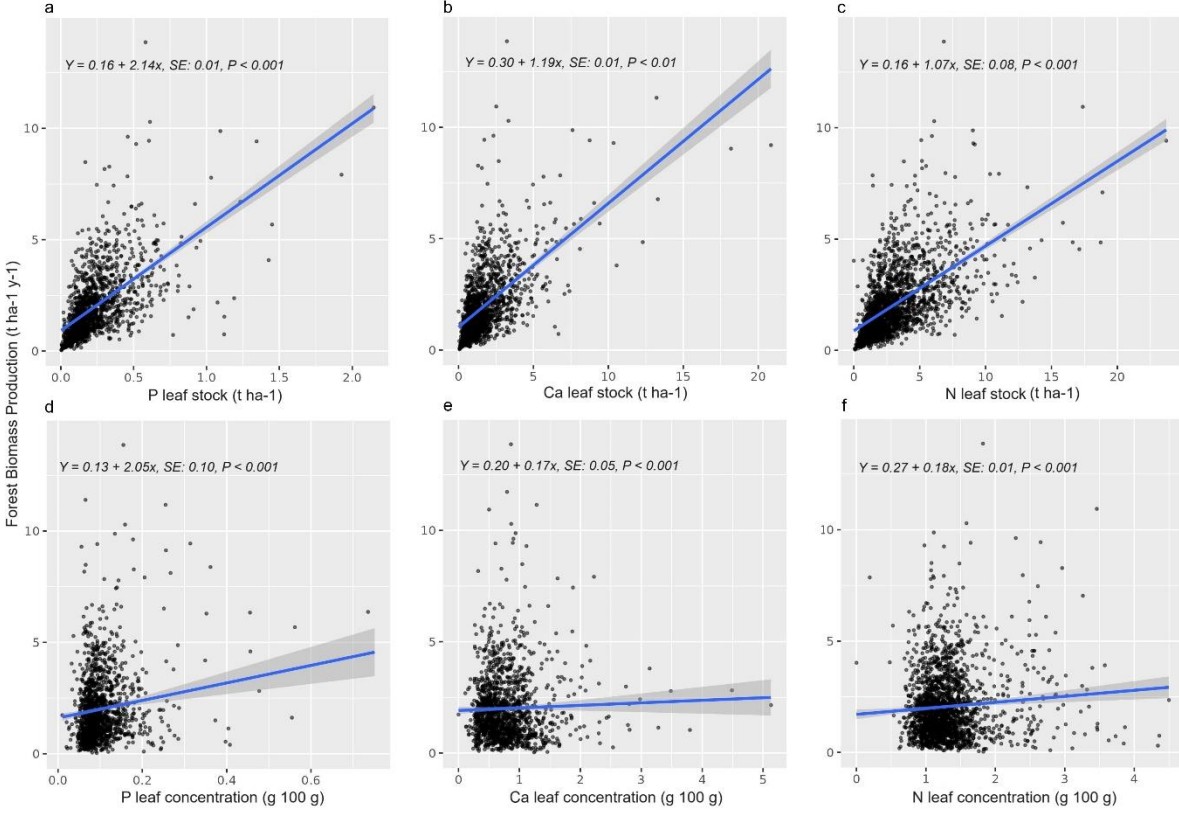


**Figure A3: Partial residuals plots showing the estimated effects of the elemental concentrations and stocks of Ca, P,**

**and N on forest biomass production.** *SE*: **Standard error.**








