# Peer review of "Optimal set of leaf and aboveground tree elements for predicting forest functioning"

_EGUsphere, 2024_

## Author Comment (AC2)

**RC2: 'Comment on egusphere-2024-2572', Emma Hauser**

In the article Optimal set of leave and whole-tree elements for predicting forest functioning the authors analyze data from over 2000 trees in the Forest Inventory of Catalonia to examine the role of tree leaf and whole tree (comprised of leaf, bark, stem, and branches) elemental composition in explaining forest biomass production and productivity rates. The authors find that, while forest age best predicts forest biomass productivity, variations in forest production are better explained by leaf nutrient stocks than whole tree nutrients or nutrient concentrations. The authors also identify which foliar elements and interactions of elements can best model forest productivity and production, highlighting the importance of N and P in forest production variables.

This study presents an important advance in our understanding of forest ecosystems. First, as the authors suggest, these results could guide forest sampling, as they indicate that foliage may be sufficient to estimate forest productivity metrics. Such information could streamline forest sampling as well as model developments reliant on subsequent data. The degree to which forest ecosystem level information can be represented from foliar data is needed and useful information, especially in efforts to estimate large scale forest nutrient demands and model ecological processes. Further, this analysis pairs data from bark, stems, branches, and leaves from many individual trees, a rare number of forest data components to have all in one place. These data could give us a better sense of how different parts of a tree contribute to whole tree nutrition, as well as C and nutrient allocation patterns. The importance of these results for understanding tree resource allocation could be described more strongly in the introduction and/or discussion, but overall this work represents an important next step in our understanding of forest ecosystems.

Authors' answer: Dear referee, thank you for your feedback above described. It reinforces our motivation to keep pursuing and widening the research presented in this study.

We added at the introduction the following sentence for reinforcing the importance of studying plant nutrient stocks for understanding allocation resources and biomass growth:

"The variability of plant nutrient stocks, particularly C, N, and P, determine how trees allocate resources between roots and aboveground organs, ultimately impacting their biomass growth (Yan et al. 2016; Li et al. 2024). Therefore, assessing effects of the tree nutrient stocks on forest biomass contributes to better understand their adaptation varying nutrient and environmental conditions (Peng et al., 2020)."

However, there are a few crucial details that are missing from this paper that challenge whether leaves alone can be sufficient to estimate ecosystem function as the authors suggest. My primary concern is that there is no representation of belowground drivers of productivity, namely root and soil processes. Roots especially are an important component of tree biomass production due to their role in nutrient uptake, as well as the relatively quick growth and turnover of fine roots. Given that root data are not a part of the database, including roots may be beyond the scope of this work, but then the authors should be more explicit throughout the manuscript that their work pertains specifically to aboveground production.

Authors' answer: thank you for raising these points. To clarify and better contextualize the limitation of the absence of root and soil data (unfortunately unavailable for our dataset) in the analysis, we proceeded as following:
- Throughout the text, figures, and tables we replaced the terminology "whole plant" by "aboveground". It reflects more appropriately the outputs of our analysis when comparing other aboveground organs with leaves and also take into account that roots weren't used in the analysis, since we didn't have such data available.

- We added a "Caveats, limitations, and implications" section at the end of the discussion that highlights the importance of belowground organs like roots and soil nutrients for biomass and elements storage and recommending their inclusion in further studies using our modelling approach, reading:

"Caveats, limitations, and implications
In this study, we bring new insights into the effects of the optimal elemental sets, compared to climate and stand age, on both forest biomass production and productivity. As practical implications for future research, our results suggest that using only data on leaf elements, especially stocks, allows us to achieve robust predictions of variations in forest biomass. Such information can contribute to decision-making by researchers and forest managers about the types of data (aboveground elements or just leaves' elements) they should prioritize collecting when assessing forest growth. Nevertheless, our presented results should also be interpreted cautiously since they might be influenced by sampling limitations and analyses conducted only on aboveground organs (barks, branches, leaves, and stems). In the data used in this study, measurements of element concentrations in different above-ground organs of trees were obtained for different numbers of individuals per species. This difference in the number of individuals may have influenced, even if subtly, the results. Besides, the biomass of belowground organs (e.g., fine and coarse roots) may account for at least 22% of the total forest biomass (Ma et al., 2021) and display important roles in nutrient uptake and storage (Gao et al., 2021; Dybzinski et al., 2024). For some Mediterranean species, belowground organs may represent up to 50% of the forest biomass (Fernández-Martínez et al., 2014). Therefore, below-ground biomass and elementomes may help explain above-ground production and productivity. The importance of roots for element stocks is also underscored by the fact that around 24% of total plant carbon is stored belowground (Ma et al., 2021). Root biomass is also influenced by climate, and thus warmer and drier climates may affect the balance between aboveground and belowground biomass allocations and element stocks (Pornon et al. 2019; Ma et al., 2021). Together with roots, soil nutrient stocks are also important drivers of forest biomass, since these stocks influence the construction of both foliage and wood components (Zarzosa et al., 2021; De Vos et al., 2015; Augusto et al., 2017;). Soil nutrient availability directly influences the nutrient stocks of aboveground organs (e.g., leaves) by driving nutrient uptake and allocation, which controls photosynthesis and biomass accumulation (Augusto et al., 2022; Wiesmeier et al., 2019). Thus, including element concentrations and stocks of roots and soil nutrients (concentrations and stocks) in statistical models may enhance the predictability of forest functioning. We suggest that future research includes belowground and soil elements in addition to elements in aboveground biomass, to allow for the comparison between the predictive performance using whole-plant elements (above and belowground) and only aboveground elements".

Further, there is no mention of the role of soil nutrients even though soil nutrients are key drivers of forest productivity. There is discussion of productivity drivers other than tree nutrients in the manuscript such as climate and stand age, but soil nutrients are not part of the analysis or in any of the introductory or conclusion text. Soil nutrients are the source of tree nutrients so it seems a little odd that these are overlooked. I don't think they need to be a central focus of the manuscript, but soil nutrients are at least worth mentioning and possibly bringing into the analyses if they are available. It would be nice to also have some details about the types of soils found in these forests and any known differences between the soils in the site description.

Authors' answer: thank you for this comment. As explained in the previous answer, we recognized the importance of soil for forest biomass in a subsection "Caveats, limitations, and implications" we added at the end of the discussion section. Unfortunately, soil nutrients were not sampled in the survey of the plot data we used in the manuscript, so we were unable to either include such variables in our modelling nor make deeper inferences or suggestions based on soil roles in our results. About information of soil types in the study sites, we added information at the "Study Area" section on the most predominant soil types across forests in Catalunya, reading now:

"This study was conducted across the northeast of the Iberian Peninsula (ca. 31,900 km2), bounded in the north by the Pyrenees and in the east by the Mediterranean Sea. We chose this region due to its heterogeneous climatic conditions associated with large ranges in altitude (i.e., 0 to > 3000 m) and distance from the sea, which together result in wide variations in mean annual temperature (from 1 °C to 28 °C) and precipitation (annual mean from 350 to >1500 mm) (Martín Vide et al., 2008). Further, the forests in this region exhibit a diverse range of soil types, predominating cambisols, fluvisols, regosols, and leptosols (Soil Atlas of Europe, 2006; ICGC, 2019), with variations in organic matter and moisture content depending on the specific forest area (Selkimäki et al., 2011). The Mediterranean climate is mostly characterized by mild winters, dry and warm summers, and a high degree of interannual variability in precipitation. Such an array of environmental conditions in the study region displays significant roles in variation in elemental allocation (e.g., N, P, K), thus influencing the nutrient stocks across forest types (Sardans and Peñuelas, 2014)."

Finally, it would be helpful to clarify and expand the carbon paragraph that is brought in at the end of the discussion. In most of the manuscript, carbon is referred to in the same way as nutrients are in the analyses. Given that C and nutrients serve different roles in the plant and that biomass is approximately 50% carbon, would the authors expect a direct relationship between biomass production and C, and would they expect this relationship to be different than that between biomass and nutrients? The difference between C and nutrients in these analyses is touched on in lines 324 -329, but I'm wondering if the authors could expand this discussion and maybe bring it into the manuscript earlier, as I wondered about this as I was reading the introduction, results and discussion.

Authors' answer: thanks for the comment. We added further information on the importance of carbon, besides other important elements (N, P), in the second paragraph of the introduction, reading:

"The multi-dimensional concentration of elements of an organism has been defined as the elementome (Peñuelas et al., 2019). Assessing the elementomes of different species allows for a better understanding of how they withstand contrasting environmental conditions, since their ecological strategies rely on different element concentrations and functional traits (Peñuelas et al., 2019; Fernández-Martínez, 2022; Reich and Oleksyn, 2004). Within plant elementomes, the importance of the concentrations of C in plants is paramount because it acts as an energy store and provides structure, representing most of the plant biomass, i.e., around 46% in leaves, 47% in stems, 45 in bark and woods, and 45% in roots. (Thomas and Martin, 2012; Ma et al., 2017). The concentrations of other elements like N and P play significant roles in plant nutrition and metabolic processes and act synergistically with C (Taiz et al. 2014). For example, N is essential for protein synthesis and chlorophyll formation, directly affecting photosynthesis and carbon fixation, while P regulates energy transfer via ATP, impacting carbon assimilation and growth (Hawkesford et al., 2012). Further, considering that the concentrations of elementomes differ across species and populations in response to environmental gradients, forest ecosystems distributed over climatic gradients are expected to vary in both their species composition and elementomes (Sardans et al., 2021; Vallicrosa et al., 2022)."

Regarding the paragraph mentioned at the end of the discussion, we expanded the section where we explain the influence of other elements in C biomass allocation, while also mentioning why a direct relationship between biomass production and C cannot be always expected. Further, in this paragraph we also mention how roots influence the balance of the relationships between C, N, P, and the effects on biomass allocation. The current paragraph reads:

"Lastly, the lower relevance of C in our average models may be partially due to its variations across distinct plant organs, e.g., the predominance of leaf and fine-root turnovers in C allocations (Yu et al., 2017). Besides, foliar nutrients, particularly P, significantly impact photosynthetic C uptake in forests, promoting variation in biomass production (Mercado et al., 2011). This leads to decreased biomass production in other organs, such as stems and barks (Jonsson et al., 2020; Ryan et al., 1997; Schoonmaker et al., 2016; Yu et al., 2017). However, although plant biomass contains around 50% carbon, its production is not directly proportionate to C availability (He et al., 2020). Changes in N and P concentrations – important elements for regulating critical metabolic processes (e.g., protein synthesis, energy transfer) – may shift C allocation to maintenance and fine-root turnover, limiting structural biomass growth in stems and barks (Bruner et al., 2013; Likulunga et al., 2022). Consequently, other plant organs may allocate less C and reduce their biomass, ultimately limiting forest biomass productivity (Bruner et al., 2013; Neumann et al., 2020). Additionally, with growing P constraint under global change scenarios, C allocation patterns are projected to become more complex, directly reducing forest biomass production (Köhler et al., 2023)."

Overall I think this is an important and useful contribution to ecosystem science, but requires a bit more specificity in the text given the data presented. In the results and conclusions it often seems like the authors are making claims that are too broad and generalized for the results (for example saying that leaf nutrients are sufficient to understand ecosystem function, when leaf nutrient stocks are specific to forest production and age matters more for productivity). It would also be great if the authors

could add some information about the belowground contributions to ecosystem production, especially in the introduction and conclusions to the manuscript. In making the claims more nuanced, the paper would highlight a specific ecological relationship that is important for guiding next steps in ecosystems research in both empirical and modeling applications. I have included more specific ideas in my line by line comments below. Authors' answer: thank you for pointing this out. To avoid overly generalizing, throughout the text we have replaced terms such as "ecosystem functioning" with "forest functioning" since our object of study was forests and not various types of ecosystems. Furthermore, when the term "forest functioning" is first explained in the Introduction, we now indicate that it refers to a specific type of functioning, namely, growth in biomass (represented in the study by production and productivity). By doing so, we further highlight the specific ecological relationship we are assessing, i.e., relationships of climate, age, and elementomes with forest biomass. Regarding the recognition of the importance of belowground components (i.e., roots, soil), we added sentences in the section "Caveats, Limitations, and Implications" previously mentioned (L. XX-XX). Further, we now also stress the importance of belowground components in the introduction, sentences reading:

"Environmental conditions influence the assembly of tree communities, thus forming different forest types across distinct environmental gradients, e.g., climate and soil variation (Chu et al., 2019; Sardans et al., 2016). Soil nutrient availability (e.g., N, P, K) directly affects tree growth and is thus a key regulator of global forest productivity and forest biomass accumulation (Batjes, 1996; Wiesmeier et al. 2019). The stocks of soil nutrients are influenced by the climatic conditions that drive water availability, temperature-dependent nutrient cycling, and soil organic matter decomposition rates (Zhang et al. 2018c; Mensah et al., 2023). Such environmental conditions encompass specific niches (e.g., climatic and soil conditions) and then drive functional adaptations of the species (e.g., morphology or physiology traits) (Lavorel et al., 2007; Augusto et al. 2017; Wang et al., 2022)."

And "Considering elements (concentrations and stocks) of the entire aboveground biomass and of leaves only may contribute to enhancing the understanding of ecosystem processes (Luo et al., 2020; Rocha et al., 2011). Forest biomass production (i.e., the overall total amount of biomass accumulated over an area in a given period) is influenced by the concentration of elements the plants store (Dar and Parthasarathy, 2022; Ullah et al., 2024). Fine roots, for example, influence tree nutrient stocks since they regulate processes like water absorption and nutrient uptake from the soil (Likulunga. et al., 2022; Zhao et al., 2022). Further, tree elemental concentrations (e.g., from aboveground organs) significantly impact ecosystem productivity (Bitomský et al., 2023; Elser et al., 2010). Therefore, elemental concentrations also contribute to forest biomass productivity—a unit of biomass (e.g., per area and year) produced per unit of standing biomass that reflects ecosystem efficiency (Margalef, 1998; Lartigue and Cebrian, 2012)."

Abstract:

Lines 19-20: The statement that analyzing only leaves is a good enough approach to study ecosystem functioning seems a little too general. Ecosystem functioning can include a lot of processes besides just productivity. In addition roots and soil nutrients were not

analyzed here, which are also likely important to productivity. I'd make this sentence a little more specific to the study, possibly "our results indicate that leaf element stocks...hinting toward leaf measurements as a critical for predicting forest productivity" (or something along those lines).

Authors' answer: thank for the suggestion, we rephrased this end of the abstract following your suggestion, reading now as:

"Hence, our results indicate that leaf element stocks are better predictors of forest biomass production than aboveground element concentrations or stocks, thus hinting toward leaf measurements as critical factors for predicting variations in forest biomassproduction."

Introduction

Line 38: In these other studies, are there different measures of ecosystem function? Is elementome in these studies specifically correlated with productivity? It seems like ecosystem function and productivity are sometimes used interchangeably in the manuscript but there are numerous functions other than productivity.

Authors' answer: Thank you for your questions. We understand your concern about how we are using the term ecosystem functioning here. We replaced in the sentence the term "ecosystem functioning" with "ecosystem productivity", since all these studies assessed productivity in varied vegetation ecosystems, not only forest. For instance, Fernández-Martínez et al., 2020 assessed productivity and production in forests (e.g., evergreen, deciduous, mixed), shrublands, savannas, grasslands, and wetlands. They found, for example, that in P-rich sites, the increase of foliar N was related to increased gross primary production (GPP). Šímová et al., 2019 assessed distinct forest ecosystems (e.g., temperate, boreal, neotropical) and found a positive relationship between net primary productivity (NPP) and leaf P in tropical forests and positive relationship between NPP and leaf N in temperate forests. Yan et al. 2023 assessed distinct natural ecosystems (forests, shrublands, meadows, steppes, grasslands) regarding their GPP (i.e., yearly and monthly amount of carbon dioxide that is converted into organic matter by plants). They found that leaf P and C concentrations, mediated by leaf area and biomass, may lead to a positive relationship with GPP. Therefore, our intention with the sentence as it stands was to emphasize the importance of leaf elementomes in an array of ecosystems and their productivity.

Line 42: I would add roots to this list.
Authors' answer: Added.

Lines 47-49: The first two sentences of this paragraph feel repetitive with the beginning of the last paragraph. Maybe these two paragraphs could be trimmed and condensed.

Authors' answer: We rephrased it the first sentences of this, and the previous paragraph mentioned, reading:

"Most studies analyzing ecosystem productivity found significant correlations with leaf elementomes"
(Here we want to introduce the focus of the paragraph on the topic of using leaves vs. whole aboveground or whole-plant elementomes to predict forest functioning [i.e., biomass productivity and production])

And

"Considering elements (concentrations and stocks) of the entire aboveground biomass and of leaves only may contribute to enhancing the understanding of ecosystem processes".

(Here we want to introduce the focus of the paragraph on the topic of using elementomes [i.e, concentrations] vs. stock to predict forest functioning [i.e., biomass productivity and production])

Line 61: Should ODs be OES?
Authors' answer: Yes. We corrected it accordingly.

Line 65: Could the authors describe more explicitly in this sentence why the environmental gradient and different forest forms are important to testing OES topics?
Authors' answer: We added additional sentences to the first two ones to explain this, now reading:

"In this study, we used a database including forest elemental composition and biomass growth in the northeast of the Iberian Peninsula. This region is a suitable model for investigating topics related to OES, as it is composed of a notable environmental gradient (e.g., wide variations in climate and altitude) that influences the formation of distinct forest types (Sardans and Peñuelas, 2014). Variations in climate, soil nutrients, and species composition lead to differences in plant stoichiometry (e.g., balance in the C, N, and P) across distinct forest types, thus affecting their growth rates and biomass accumulation (Sardans and Peñuelas, 2014; Shi et al., 2016). Therefore, environmental gradients, such as the cited study region, allows for more robust assessments of general trends in the influence of OES on forest biomass growth."

Line 69: What do the authors mean by 'departed from' here? A rewording might make the intent clearer.
Authors' answer: We rephrased it, now reading: "Related to these questions, we established three central hypotheses"

Line 73: a concluding sentence that wraps up the introduction stating why these findings will be important would be nice here.
Authors' answer: we added a final sentence, reading:

"Answering the questions above can contribute significantly to enhancing the knowledge about the role of plant elementomes in forest growth, while providing practical insights for researchers and managers on which type of elemental data (e.g., aboveground elements or just leaves' elements) to collect and assess."

Materials and Methods

Study area—it would be great to be a little more specific about why this study area is useful/chosen. Does climate diversity suggest there will also be elementome diversity? Will this allow the authors to test different effects of climate vs. elementome? The authors

do have some text relevant to this in lines 35-36, but it might be nice to add that here as well (or maybe move some of that here) to make it clearer why these sites were chosen.
Authors' answer: We explained the reason for choosing this study area in the sentences we added in the final paragraph of introduction (cited in above comments), reading:

"In this study, we used a database including forest elemental composition and biomass growth in the northeast of the Iberian Peninsula. This region is a suitable model for investigating topics related to OES, as it is composed of a notable environmental gradient (e.g., wide variations in climate and altitude) that influences the formation of distinct forest types (Sardans and Peñuelas, 2014). Variations in climate, soil nutrients, and species composition lead to differences in plant stoichiometry (e.g., balance in the C, N, and P) across distinct forest types, thus affecting their growth rates and biomass accumulation (Sardans and Peñuelas, 2014; Shi et al., 2016). Therefore, environmental gradients, such as the cited study region, allows for more robust assessments of general trends in the influence of OES on forest biomass growth."

Furthermore, in the "Study Area" section in Methods, we added the following sentences to reinforce the characteristics of the study region that led us to choose it:

"Further, the forests in this region exhibit a diverse range of soil types, predominating cambisols, fluvisols, regosols, and leptosols (Soil Atlas of Europe, 2006; ICGC, 2019), with variations in organic matter and moisture content depending on the specific forest area (Selkimäki et al., 2011). The Mediterranean climate is mostly characterized by mild winters, dry and warm summers, and a high degree of interannual variability in precipitation. Such an array of environmental conditions in the study region displays significant roles in variation in elemental allocation (e.g., N, P, K), thus influencing the nutrient stocks across forest types (Sardans and Peñuelas, 2014)."

Line 113: might rephrase "5 to 5" as "each 5 cm increment" to make clearer what is meant here.
Authors' answer: We corrected as suggested.

Line 115: Since root data are not available, it would be more accurate to say "aboveground productivity" throughout the manuscript to make it clear that that's what is being examined here.
Authors' answer: We replaced it with "aboveground organs". Further, at the end of this same paragraph we added a sentence just to ensure the reader has in mind we only used aboveground biomass in our analysis, reading:

"Therefore, we emphasize that in our study, forest biomass production and productivity were measured considering only above-ground tree components."

Besides, throughout the manuscript we replaced all terms "whole-plant" and "whole-plant elementomes" (previously used) by "aboveground organs" and "aboveground elementomes", respectively.

Line 118: It might be worth it to set the equations out in a separate line rather than having them embedded within the paragraph. That can make some of the math a little easier to follow.
Authors' answer: Yes, we set it in a separate line as suggested.

Lines 119-120: here and throughout I found the use of production vs. productivity somewhat confusing. I understand the difference, and these may be the established conventions in which case this can be disregarded, but if it would be possible to rename one of the terms so they are more distinct, it would make it easier to follow which is being discussed later in the paper.
Authors' answer: We added new sentences at the fourth paragraph (current lines 48-53) of the introduction section for ensuring the clarity between the two terms, reading:

"Considering elements (concentrations and stocks) of the entire aboveground biomass and of leaves only may contribute to enhancing the understanding of ecosystem processes (Luo et al., 2020; Rocha et al., 2011). Forest biomass production (i.e., the overall total amount of biomass accumulated over an area in a given period) is influenced by the concentration of elements the plants store (Dar and Parthasarathy, 2022; Ullah et al., 2024). Fine roots, for example, influence tree nutrient stocks since they regulate processes like water absorption and nutrient uptake from the soil (Likulunga. et al., 2022; Zhao et al., 2022). Further, tree elemental concentrations (e.g., from aboveground organs) significantly impact ecosystem productivity (Bitomský et al., 2023; Elser et al., 2010). Therefore, elemental concentrations also contribute to forest biomass productivity—a unit of biomass (e.g., per area and year) produced per unit of standing biomass that reflects ecosystem efficiency (Margalef, 1998; Lartigue and Cebrian, 2012)."

Line 141: It would be a little clearer to put 'generalized additive mixed models' first and then put (GAMMs) in parentheses.
Authors' answer: We rephrased as suggested. Thank you.

Results

Line 191: Here 'forest functioning' is used a little too broadly. I'd suggest using productivity or production since that is what was measured.
Authors' answer: We replaced "forest functioning" with "forest biomass production and productivity".

Line 191-193: I found these sentences hard to follow. It seems like there are 2 models presented, one in each sentence, but they are each described as the 'best model.' Is the second sentence for productivity and that word has been omitted from the sentence?
Authors' answer: For better clarifying it, we rephrased the second sentence of these two, now both read:
"We found that the best model of forest biomass production using leaf element stocks as predictors explained 58% of the variance and had nine variables: C, Ca, K, Mg, N, P, C×N, C×P, and N×P (Fig. 1a). The second-best model explained 28% of the variance of forest biomass production (Fig. 1a) had three aboveground element stocks as predictors (C, N, and C×N)."

Line 200: Nothing is mentioned about climate or age in the first paragraph but figure 1 seems to suggest that climate/age explains production best so might be worth mentioning that in this first paragraph.

Authors' answer: We added a new sentence for mentioning it. Further, we rephrased the other two sentences and merged into one for simplicity and clarity. These changes now read:

"Forest biomass productivity was best predicted by the model with climate and stand age as predictors (Fig. 1c, d). Secondarily, between leaf elementomes (Ca, K, and N) and aboveground elementomes (K), the first ones were the best predictors of forest biomass productivity (Fig. 1c; 28% of variance explained)."

Discussion

Line 289-294: This may just be a wording issue, but it seems like these sentences are contradictory. The sentence starting "We found a possible effect..." suggests that less moisture caused plants to retain more nutrients in leaves to cope with drought. The sentence starting in line 293, "Therefore, our observed..." suggests that high precipitation coincides with high foliar nutrient storage, so it is unclear whether more nutrients are stored with more or less water.

Authors' answer: we rephrased the last sentence for adding more clarity. Now this part of the paragraph reads:

"We found a positive effect of precipitation in the driest quarters, N and P, on forest biomass production. Since, during summer, most of the territory addressed in this study coincides with high temperatures and marked water stress (Martín Vide et al., 2008), plants may invest in a strategy of retaining larger foliar nutrient reserves to cope with drought (Waring, 1987.; Gessler et al., 2017). Increased precipitation might enhance the foliar nutrients stored in drier periods, thus contributing positively to aboveground biomass production (Fernández-Martínez et al., 2017; Lie et al., 2018; Roa-Fuentes et al., 2012). In our study region high water availability (e.g., precipitation) correlates positively with mineralization, which enhances the nutrient availability to trees and contribute to increasing their biomass (Sardans et al., 2008)."

Line 325: It would be nice to restate more specifically here what the decrease in forest biomass is that the authors refer to.

Authors' answer: We rephrased it for clarity, reading:

"Finally, the smaller importance of C compared to other elements in our average models might also partially explain the decrease in forest aboveground biomass productivity. Variations in both aboveground and belowground biomass might be influenced, for example, by the predominance of leaf and fine-root turnovers in carbon allocations compared to other plant parts (Yu et al., 2017)."

Conclusion

Lines 343-345: The authors point out that productivity was not driven by nutrients but stand age, then suggest that focusing on leaf elements is sufficient for understanding variations in forest biomass. This seems a little misleading–more weight should be given to

the fact that other variables besides leaf nutrients were primary drivers and that root and soil nutrients were not considered as a part of this analysis.

Authors' answer: we toned down the statement by rephrasing the last sentence, now reading:

"Altogether, our results indicate that leaf element stocks are critical predictors of forest biomass production".

Regarding the recognition of the role of other factors like soil and roots, we already addressed it the section "Caveats, limitations, and implications", described in answers to previous comments.

---

## Author Response (AR1)

**RC1: 'Comment on egusphere-2024-2572', Helena Vallicrosa**

In this study, Diniz et al. are using elemental composition and biomass information from the Ecological and Forest Inventory of Catalonia (trees) to determine the best variables to determine forest functioning. The study is relevant and interesting, and the methodology seems sound. Nonetheless, the manuscript requires attention in some aspects before being suitable for publication:

Authors' answer: Dear referee, thank you for your feedback on our manuscript and the contributions for your comments and suggestions. Thank you very much also for the references you shared. We really appreciate it and acknowledge for the time you devoted to conduct this review. Below, we provide answers to each of your comments.

General comments:

My main concern is the use of "whole-tree" terminology to describe only the aboveground section of the tree. I acknowledge the value of including information coming from leaves but also branches, stems and bark, which is rare to find in a dataset that combines biomass and elemental composition information. Nonetheless, disregarding roots in the conception of "whole-tree" is misleading. Belowground biomass (coarse and fine roots) represents at least 22% of the total forest biomass in forests (Ma et al., 2021), and play key roles in tree growth, stand development and nutrient uptake (Gao et al., 2021). Therefore, including the role of roots in a similar analysis would be extremely valuable and interesting. For instance, a recent study emphasized the importance of accounting for both leaves and fine roots in the calculation of plant nutrient uptakes, where stems could be inferred or gap-filled from inventories (Dybzinski et al., 2024). In conclusion, we all seem to agree that based on the evidence provided by Diniz et al. and other studies, stem-related tissues are not as relevant for forest functioning determination, but below-ground information is desired. Thus, I recommend changing the terminology used in this study to above-ground elements instead of whole-tree.

Authors' answer: Thank you for pointing this out. Indeed, the terminology aboveground is more appropriate to our context since our analysis did not include roots data. We replaced 'whole-plant' by 'aboveground' throughout the text and in the tables and figures captions.

Related to the above, it would be desirable to include a paragraph or section in the discussion disclosing whether the non-inclusion of below-ground tissues can affect the conclusions of the study and encourage the use of below-ground information in future studies. Also, other "disclaimers" regarding the methodology could be fleshed out. For instance, could the low predictive power of "whole-plant" related analysis be attributed to the aggregation of error while sampling production and elements of branches, stems and bark?

Authors' answer: We added a "Caveats, limitations, and implications" section at the end of the discussion and included sentences recognizing the importance of belowground organs like roots for biomass and elements storage and recommending their inclusion in further studies using our modelling approach, (lines 374-399), reading:

"Caveats, limitations, and implications:
In this study, we bring new insights into the effects of the optimal elemental sets, compared to climate and stand age, on both forest biomass production and productivity. As practical implications for future research, our results suggest that using only data on leaf elements, especially stocks, allows us to achieve robust predictions of variations in forest biomass. Such information can contribute to decision-making by researchers and forest managers about the types of data (aboveground elements or only leaves' elements)

they should prioritize collecting when assessing forest growth. Nevertheless, our presented results might be influenced by sampling limitations and analyses conducted only on aboveground organs (barks, branches, leaves, and stems). In the data used in this study, measurements of element concentrations in different above-ground organs of trees were obtained for various numbers of individuals per species. This difference in the number of individuals may have influenced, even if subtly, the results. Besides, the biomass of belowground organs (e.g., fine and coarse roots) may account for at least 22% of the total forest biomass (Ma et al., 2021) and display important roles in nutrient uptake and storage (Gao et al., 2021; Dybzinski et al., 2024). For some Mediterranean species, belowground organs may represent up to 50% of the forest biomass (Fernández-Martínez et al., 2014). Therefore, below-ground biomass and elementomes may help explain above-ground production and productivity. The importance of roots for element stocks is also underscored by the fact that around 24% of total plant carbon is stored belowground (Ma et al., 2021). Root biomass is also influenced by climatic factors such as temperature, thus leading us to expect that future changes driven by warmer and drier climates will affect the balance between aboveground and belowground biomass allocations and element stocks (Pornon et al., 2019; Ma et al., 2021). Alongside roots, soil nutrient stocks are also important contributors to forest biomass, since these stocks influence the construction of foliage and wood components (Zarzosa et al., 2021; De Vos et al., 2015; Augusto et al., 2017). Soil nutrient availability directly influences aboveground organs (e.g., leaves) nutrient stocks by driving nutrient uptake and allocation, which controls photosynthesis and biomass accumulation (Augusto et al., 2022; Wiesmeier et al., 2019). Thus, including element concentrations and stocks of roots and soil nutrients (concentrations and stocks) in statistical models may enhance the predictability of forest functioning. We suggest that future research includes belowground and soil elements in addition to elements in aboveground biomass, to allow for the comparison between the predictive performance using whole-plant elements (above and belowground) and only aboveground elements.

Using the methodology and data followed in this paper, I believe it would be interesting also to calculate the predictive power of single elements and relative stocks. What is the element describing ecosystem functioning the most? Does it correspond with the most sampled (normally N)? Could this be a reason to start sampling other elements more? Even though it is not the same information, could maybe this information be somehow inferred from the information displayed in Figure 3a?

Authors' answer: Thank you for your comment. The information on the importance of predictors based on individual elements (concentrations and stocks) in contributing to the performance of models in predicting forest functioning can be inferred from the outputs of the model averages presented in Figures 3, 4 and A3. In the methods section (lines 209-215) we explain that, reading:

"Finally, to obtain a reliable overview of which were the most important variables (e.g., elements concentration and stocks) for explaining forest functioning, we performed model averaging for models with ΔAIC < 4 using the function "model.avg" of the "MuMIn" package (Bartoń, 2023) in R 4.3.3. We used the argument "beta=TRUE" to standardize the coefficients, allowing for a comparison of the relative importance of each predictor variable in the average models. Model averaging computes an average model output from the estimates of a set of models and weights their relative importance by their AIC (Burnham and Anderson, 2002). Therefore, this approach allowed us to obtain information on the importance of predictor variables extracted from the best model subsets (i.e., ΔAIC < 4)."

Thus, for example, the results in figure A3 show the elements (Ca, N, P) that most affect forest functioning, their coefficients having been based on the average of several well-ranked models (delta < 4).

Further, we added a sentence in the results section (lines 258-259) to summarize it more explicitly. Reading:
"The information contained in the figures 3, 4 and A3 outline the importance of individual elements (concentrations and stocks) in contributing to the performance of models in predicting forest functioning."

Specific comments:

Line 11: What is the difference between forest production and productivity? It would be desired to describe the difference between productivity and production somewhere in the introduction. Both are concepts used throughout the paper that can be easily confused.
Authors' answer: We added new sentences in the fourth paragraph (lines 57-65), reading:

"Considering elements (concentrations and stocks) of the entire aboveground biomass and leaves only may contribute to enhancing the understanding of ecosystem processes (Luo et al., 2020; Rocha et al., 2011). Forest biomass production (i.e., the overall total amount of biomass accumulated over an area in a given period) is influenced by the concentration of elements the plants store (Dar and Parthasarathy, 2022; Ullah et al., 2024). Fine roots, for example, influence tree nutrient stocks since they regulate processes like water absorption and nutrient uptake from the soil (Likulunga. et al., 2022; Zhao et al., 2022). Further, tree elemental concentrations (e.g., from aboveground organs) significantly impact ecosystem productivity (Bitomský et al., 2023; Elser et al., 2010). Therefore, elemental concentrations and stocks also contribute to forest biomass productivity—a unit of biomass (e.g., per area and year) produced per unit of standing biomass that reflects ecosystem efficiency (Margalef, 1998; Lartigue and Cebrian, 2012)."

Line 65: Any other gradient other than forest formations? Altitude, climates…? You mention this in the methods section and including it here would help emphasize the value of the database.
Authors' answer: We added in the sentence (current line 68-70) the climate and altitude as part of a notable environmental gradient that results also in distinct forest formations, reading:
"This region is a suitable model to investigate topics related to OES, as it is composed of a notable environmental gradient (e.g., wide variations in climate and altitude) that influences the formation of distinct forest types."

Line 70-72: I would rewrite it as such (or something similar): element stocks better explain functioning than elementomes, as the former incorporates the effect of growth, encompassing factors such as age and hidden limitations in forest functioning;
Authors' answer: Thank you. We incorporated your suggestion in the sentence (current lines 76-78), reading:
"H2: element stocks better explain functioning than elementomes, as the former incorporates the effect of growth, while also encompasses effects of factors such as age and hidden limitations (e.g., carbon saturation, nutrient limitation), in forest functioning age"

Methods: What is the time scale of the sampling? When were the samples collected?
Authors' answer: We added the period (years) of the sampling (lines 124-125), reading:

"We used the Ecological and Forest Inventory of Catalonia (IEFC) database, originally sampled in the period 1989-1996 (Gracia et al., 2004) (http://www.creaf.uab.es/iefc)."

Technical corrections:

Line 60: Please, join the two parenthesis sections in one
Authors' answer: We corrected it (lines 79-80).

Line 61: I don't think ODs has been defined before. Please define.
Authors' answer: It was a typo. We wanted mean "OES". So, we replaced ODs for OES (Line 80).

Line 64: OES was previously defined. No need to define it again. Same in line 160.
Authors' answer: We deleted the repetition of definition (lines 83, ).

Line 65-73: It would help clarity to number the questions and make them match with the hypothesis.
Authors' answer: We added numbering to the questions and related them to the hypotheses (lines 88-100), reading:
"We aimed to answer four questions: Q1-Are the aboveground elements (elementomes and stocks) better predictors of forest functioning (biomass production and productivity) than only leaf elements? Q2-Do element stocks better explain forest functioning than elementomes? Q3-Do element stocks and elementomes (leaf and aboveground) explain better forest functioning than environmental factors and stand age? Q4-What is the OES that best predicts forest functioning? Related to these questions, we established three central hypotheses.: H1: Aboveground elements (elementomes and stocks) are better predictors of forest functioning (biomass production and productivity) than only leaf elements (Q1); H2: Element stocks better explain functioning than elementomes, as the former incorporates the effect of growth, while also encompasses effects of factors such as age and hidden limitations (e.g., carbon saturation, nutrient limitation), in forest functioning (Q2, Q3); H3: OES effects in forest biomass production and productivity models are greater in models using whole organisms than leaf elementomes (Q4). Answering the questions above can contribute significantly to enhancing the knowledge about the role of plant elementomes in forest growth while providing practical insights for researchers and managers on which type of elemental data (e.g., aboveground elements or only leaves' elements) to collect and assess."

Ma, H., Mo, L., Crowther, T.W. et al. The global distribution and environmental drivers of aboveground versus belowground plant biomass. Nat Ecol Evol 5, 1110–1122 (2021). https://doi.org/10.1038/s41559-021-01485-1

Guoqiang Gao, Zhi Liu, Yan Wang, Siyuan Wang, Cunyong Ju, Jiacun Gu. Tamm Review: Fine root biomass in the organic (O) horizon in forest ecosystems: Global patterns and controlling factors, Forest Ecology and Management, 491,2021, https://doi.org/10.1016/j.foreco.2021.119208.

Dybzinski, R., Segal, E., McCormack, M.L. et al. Calculating Nitrogen Uptake Rates in Forests: Which Components Can Be Omitted, Simplified, or Taken from Trait Databases and Which Must Be Measured In Situ?. Ecosystems 27, 739–763 (2024). https://doi.org/10.1007/s10021-024-00919-8

**RC2: 'Comment on egusphere-2024-2572', Emma Hauser**

In the article Optimal set of leave and whole-tree elements for predicting forest functioning the authors analyze data from over 2000 trees in the Forest Inventory of Catalonia to examine the role of tree leaf and whole tree (comprised of leaf, bark, stem, and branches) elemental composition in explaining forest biomass production and productivity rates. The authors find that, while forest age best predicts forest biomass productivity, variations in forest production are better explained by leaf nutrient stocks than whole tree nutrients or nutrient concentrations. The authors also identify which foliar elements and interactions of elements can best model forest productivity and production, highlighting the importance of N and P in forest production variables.

This study presents an important advance in our understanding of forest ecosystems. First, as the authors suggest, these results could guide forest sampling, as they indicate that foliage may be sufficient to estimate forest productivity metrics. Such information could streamline forest sampling as well as model developments reliant on subsequent data. The degree to which forest ecosystem level information can be represented from foliar data is needed and useful information, especially in efforts to estimate large scale forest nutrient demands and model ecological processes. Further, this analysis pairs data from bark, stems, branches, and leaves from many individual trees, a rare number of forest data components to have all in one place. These data could give us a better sense of how different parts of a tree contribute to whole tree nutrition, as well as C and nutrient allocation patterns. The importance of these results for understanding tree resource allocation could be described more strongly in the introduction and/or discussion, but overall this work represents an important next step in our understanding of forest ecosystems.

Authors' answer: Dear referee, thank you for your feedback above described. It reinforces our motivation to keep pursuing and widening the research presented in this study.
We added at the introduction the following sentence for reinforcing the importance of studying plant nutrient stocks for understanding allocation resources and biomass growth (lines 68-72):
"The variability of plant nutrient stocks, particularly C, N, and P, determine how trees allocate resources between roots and aboveground organs, ultimately impacting their biomass growth (Yan et al. 2016; Li et al. 2024). Therefore, assessing effects of the tree nutrient stocks on forest biomass contributes to better understand their adaptation varying nutrient and environmental conditions (Peng et al., 2020)."

However, there are a few crucial details that are missing from this paper that challenge whether leaves alone can be sufficient to estimate ecosystem function as the authors suggest. My primary concern is that there is no representation of belowground drivers of productivity, namely root and soil processes. Roots especially are an important component of tree biomass production due to their role in nutrient uptake, as well as the relatively quick growth and turnover of fine roots. Given that root data are not a part of the database, including roots may be beyond the scope of this work, but then the authors should be more explicit throughout the manuscript that their work pertains specifically to aboveground production.

Authors' answer: thank you for raising these points. To clarify and better contextualize the limitation of the absence of root and soil data (unfortunately unavailable for our dataset) in the analysis, we proceeded as following:

- Throughout the text, figures, and tables we replaced the terminology "whole plant" by "aboveground". It reflects more appropriately the outputs of our analysis when comparing other aboveground organs with leaves and also take into account that roots weren't used in the analysis, since we didn't have such data available.

- We added a "Caveats, limitations, and implications" section at the end of the discussion that highlights the importance of belowground organs like roots and soil nutrients for biomass and elements storage and recommending their inclusion in further studies using our modelling approach, reading (lines 374-399):

"Caveats, limitations, and implications
In this study, we bring new insights into the effects of the optimal elemental sets, compared to climate and stand age, on both forest biomass production and productivity. As practical implications for future research, our results suggest that using only data on leaf elements, especially stocks, allows us to achieve robust predictions of variations in forest biomass. Such information can contribute to decision-making by researchers and forest managers about the types of data (aboveground elements or just leaves' elements) they should prioritize collecting when assessing forest growth. Nevertheless, our presented results should also be interpreted cautiously since they might be influenced by sampling limitations and analyses conducted only on aboveground organs (barks, branches, leaves, and stems). In the data used in this study, measurements of element concentrations in different above-ground organs of trees were obtained for different numbers of individuals per species. This difference in the number of individuals may have influenced, even if subtly, the results. Besides, the biomass of belowground organs (e.g., fine and coarse roots) may account for at least 22% of the total forest biomass (Ma et al., 2021) and display important roles in nutrient uptake and storage (Gao et al., 2021; Dybzinski et al., 2024). For some Mediterranean species, belowground organs may represent up to 50% of the forest biomass (Fernández-Martínez et al., 2014). Therefore, below-ground biomass and elementomes may help explain above-ground production and productivity. The importance of roots for element stocks is also underscored by the fact that around 24% of total plant carbon is stored belowground (Ma et al., 2021). Root biomass is also influenced by climate, and thus warmer and drier climates may affect the balance between aboveground and belowground biomass allocations and element stocks (Pornon et al. 2019; Ma et al., 2021). Together with roots, soil nutrient stocks are also important drivers of forest biomass, since these stocks influence the construction of both foliage and wood components (Zarzosa et al., 2021; De Vos et al., 2015; Augusto et al., 2017;). Soil nutrient availability directly influences the nutrient stocks of aboveground

organs (e.g., leaves) by driving nutrient uptake and allocation, which controls photosynthesis and biomass accumulation (Augusto et al., 2022; Wiesmeier et al., 2019). Thus, including element concentrations and stocks of roots and soil nutrients (concentrations and stocks) in statistical models may enhance the predictability of forest functioning. We suggest that future research includes belowground and soil elements in addition to elements in aboveground biomass, to allow for the comparison between the predictive performance using whole-plant elements (above and belowground) and only aboveground elements".

Further, there is no mention of the role of soil nutrients even though soil nutrients are key drivers of forest productivity. There is discussion of productivity drivers other than tree nutrients in the manuscript such as climate and stand age, but soil nutrients are not part of the analysis or in any of the introductory or conclusion text. Soil nutrients are the source of tree nutrients so it seems a little odd that these are overlooked. I don't think they need to be a central focus of the manuscript, but soil nutrients are at least worth mentioning and possibly bringing into the analyses if they are available. It would be nice to also have some details about the types of soils found in these forests and any known differences between the soils in the site description.

Authors' answer: thank you for this comment. As explained in the previous answer, we recognized the importance of soil for forest biomass in a subsection "Caveats, limitations, and implications" (lines 374-399) we added at the end of the discussion section. Unfortunately, soil nutrients were not sampled in the survey of the plot data we used in the manuscript, so we were unable to either include such variables in our modelling nor make deeper inferences or suggestions based on soil roles in our results. About information of soil types in the study sites, we added information at the "Study Area" section on the most predominant soil types across forests in Catalunya, reading now (lines 106-115):

"This study was conducted across the northeast of the Iberian Peninsula (ca. 31,900 km2), bounded in the north by the Pyrenees and in the east by the Mediterranean Sea. We chose this region due to its heterogeneous climatic conditions associated with large ranges in altitude (i.e., 0 to > 3000 m) and distance from the sea, which together result in wide variations in mean annual temperature (from 1 °C to 28 °C) and precipitation (annual mean from 350 to >1500 mm) (Martín Vide et al., 2008). Further, the forests in this region exhibit a diverse range of soil types, predominating cambisols, fluvisols, regosols, and leptosols (Soil Atlas of Europe, 2006; ICGC, 2019), with variations in organic matter and moisture content depending on the specific forest area (Selkimäki et al., 2011). The Mediterranean climate is mostly characterized by mild winters, dry and warm summers, and a high degree of interannual variability in precipitation. Such an array of environmental conditions in the study region displays significant roles in variation in elemental allocation (e.g., N, P, K), thus influencing the nutrient stocks across forest types (Sardans and Peñuelas, 2014)."

Finally, it would be helpful to clarify and expand the carbon paragraph that is brought in at the end of the discussion. In most of the manuscript, carbon is referred to in the same way as nutrients are in the analyses. Given that C and nutrients serve different roles in the plant and that biomass is approximately 50% carbon, would the authors expect a direct

relationship between biomass production and C, and would they expect this relationship to be different than that between biomass and nutrients? The difference between C and nutrients in these analyses is touched on in lines 324 -329, but I'm wondering if the authors could expand this discussion and maybe bring it into the manuscript earlier, as I wondered about this as I was reading the introduction, results and discussion.
Authors' answer: thanks for the comment. We added further information on the importance of carbon, besides other important elements (N, P), in the second paragraph of the introduction, reading (see lines 36-47):

"The multi-dimensional concentration of elements of an organism has been defined as the elementome (Peñuelas et al., 2019). Assessing the elementomes of different species allows for a better understanding of how they withstand contrasting environmental conditions since their ecological strategies rely on different element concentrations and functional traits (Peñuelas et al., 2019; Fernández-Martínez, 2022; Reich and Oleksyn, 2004). Within plant elementomes, the importance of the concentrations of C in plants is paramount because it acts as an energy store and provides structure, representing most of the plant biomass, i.e., around 46% in leaves, 47% in stems, 45% in bark and woods, and 45% in roots. (Thomas and Martin, 2012; Ma et al., 2017). The concentrations of other elements like N and P play significant roles in plant nutrition and metabolic processes and act synergistically with C (Taiz et al. 2014). For example, N is essential for protein synthesis and chlorophyll formation, directly affecting photosynthesis and carbon fixation, while P regulates energy transfer via ATP, impacting carbon assimilation and growth (Hawkesford et al., 2012). Further, considering that the concentrations of elementomes differ across species and populations in response to environmental gradients, forest ecosystems distributed over climatic gradients are expected to vary in both their species composition and elementomes (Sardans et al., 2021; Vallicrosa et al., 2022)."

Regarding the paragraph mentioned at the end of the discussion, we expanded the section where we explain the influence of other elements in C biomass allocation, while also mentioning why a direct relationship between biomass production and C cannot be always expected. Further, in this paragraph we also mention how roots influence the balance of the relationships between C, N, P, and the effects on biomass allocation. The current paragraph reads (lines 361-372).

[revised manuscript text omitted]

Abstract:

Lines 19-20: The statement that analyzing only leaves is a good enough approach to study ecosystem functioning seems a little too general. Ecosystem functioning can include a lot of processes besides just productivity. In addition roots and soil nutrients were not analyzed here, which are also likely important to productivity. I'd make this sentence a little more specific to the study, possibly "our results indicate that leaf element stocks...hinting toward leaf measurements as a critical for predicting forest productivity" (or something along those lines).

Authors' answer: thank for the suggestion, we rephrased this end of the abstract following your suggestion, reading now as (L.XX-XX):

"Hence, our results indicate that leaf element stocks are better predictors of forest biomass production than aboveground element concentrations or stocks, thus hinting toward leaf measurements as critical factors for predicting variations in forest biomass production."

Introduction

Line 38: In these other studies, are there different measures of ecosystem function? Is elementome in these studies specifically correlated with productivity? It seems like ecosystem function and productivity are sometimes used interchangeably in the manuscript but there are numerous functions other than productivity.

Authors' answer: Thank you for your questions. We understand your concern about how we are using the term ecosystem functioning here. We replaced in the sentence the term "ecosystem functioning" with "ecosystem productivity", since all these studies assessed productivity in varied vegetation ecosystems, not only forest. For instance, Fernández-Martínez et al., 2020 assessed productivity and production in forests (e.g., evergreen, deciduous, mixed), shrublands, savannas, grasslands, and wetlands. They found, for example, that in P-rich sites, the increase of foliar N was related to increased gross primary production (GPP). Šímová et al., 2019 assessed distinct forest ecosystems (e.g., temperate, boreal, neotropical) and found a positive relationship between net primary productivity (NPP) and leaf P in tropical forests and positive relationship between NPP and leaf N in temperate forests. Yan et al. 2023 assessed distinct natural ecosystems (forests, shrublands, meadows, steppes, grasslands) regarding their GPP (i.e., yearly and monthly amount of carbon dioxide that is converted into organic matter by plants). They found that leaf P and C concentrations, mediated by leaf area and biomass, may lead to a positive relationship with GPP. Therefore, our intention with the sentence as it stands was to emphasize the importance of leaf elementomes in an array of ecosystems and their productivity.

Line 42: I would add roots to this list.
Authors' answer: Added (lines 49, 52-53).

Lines 47-49: The first two sentences of this paragraph feel repetitive with the beginning of the last paragraph. Maybe these two paragraphs could be trimmed and condensed.
Authors' answer: We rephrased it the first sentences of this, and the previous paragraph mentioned, reading:

"Most studies analyzing ecosystem productivity found significant correlations with leaf elementomes" (line 48)
(Here we want to introduce the focus of the paragraph on the topic of using leaves vs. whole aboveground or whole-plant elementomes to predict forest functioning [i.e., biomass productivity and production])

And
"Considering elements (concentrations and stocks) of the entire aboveground biomass and of leaves only may contribute to enhancing the understanding of ecosystem processes". (line 57-58).
(Here we want to introduce the focus of the paragraph on the topic of using elementomes [i.e, concentrations] vs. stock to predict forest functioning [i.e., biomass productivity and production])

Line 61: Should ODs be OES?
Authors' answer: Yes. We corrected it accordingly (line 80).

Line 65: Could the authors describe more explicitly in this sentence why the environmental gradient and different forest forms are important to testing OES topics?
Authors' answer: We added additional sentences to the first two ones to explain this, now reading (lines 82-88):

"In this study, we used a database including forest elemental composition and biomass growth in the northeast of the Iberian Peninsula. This region is a suitable model for investigating topics related to OES, as it is composed of a notable environmental gradient (e.g., wide variations in climate and altitude) that influences the formation of distinct forest types (Sardans and Peñuelas, 2014). Variations in climate, soil nutrients, and species composition lead to differences in plant stoichiometry (e.g., balance in the C, N, and P) across distinct forest types, thus affecting their growth rates and biomass accumulation (Sardans and Peñuelas, 2014; Shi et al., 2016). Therefore, environmental gradients, such as the cited study region, allows for more robust assessments of general trends in the influence of OES on forest biomass growth."

Line 69: What do the authors mean by 'departed from' here? A rewording might make the intent clearer.
Authors' answer: We rephrased it, now reading: "Related to these questions, we established three central hypotheses" (lines 92-93).

Line 73: a concluding sentence that wraps up the introduction stating why these findings will be important would be nice here.
Authors' answer: we added a final sentence, reading (lines 98-100):

"Answering the questions above can contribute significantly to enhancing the knowledge about the role of plant elementomes in forest growth while providing practical insights for researchers and managers on which type of elemental data (e.g., aboveground elements or only leaves' elements) to collect and assess."

Materials and Methods

Study area—it would be great to be a little more specific about why this study area is useful/chosen. Does climate diversity suggest there will also be elementome diversity? Will this allow the authors to test different effects of climate vs. elementome? The authors do have some text relevant to this in lines 35-36, but it might be nice to add that here as well (or maybe move some of that here) to make it clearer why these sites were chosen. Authors' answer: We explained the reason for choosing this study area in the sentences we added in the final paragraph of introduction (cited in above comments), reading:

"In this study, we used a database including forest elemental composition and biomass growth in the northeast of the Iberian Peninsula. This region is a suitable model for investigating topics related to OES, as it is composed of a notable environmental gradient (e.g., wide variations in climate and altitude) that influences the formation of distinct forest types (Sardans and Peñuelas, 2014). Variations in climate, soil nutrients, and species composition lead to differences in plant stoichiometry (e.g., balance in the C, N, and P) across distinct forest types, thus affecting their growth rates and biomass accumulation (Sardans and Peñuelas, 2014; Shi et al., 2016). Therefore, environmental gradients, such as the cited study region, allows for more robust assessments of general trends in the influence of OES on forest biomass growth."

Furthermore, in the "Study Area" section in Methods, we added the following sentences to reinforce the characteristics of the study region that led us to choose it:

"Further, the forests in this region exhibit a diverse range of soil types, predominating cambisols, fluvisols, regosols, and leptosols (Soil Atlas of Europe, 2006; ICGC, 2019), with variations in organic matter and moisture content depending on the specific forest area (Selkimäki et al., 2011). The Mediterranean climate is mostly characterized by mild winters, dry and warm summers, and a high degree of interannual variability in precipitation. Such an array of environmental conditions in the study region displays significant roles in variation in elemental allocation (e.g., N, P, K), thus influencing the nutrient stocks across forest types (Sardans and Peñuelas, 2014)."

Line 113:  might rephrase "5 to 5" as "each 5 cm increment" to make clearer what is meant here.
Authors' answer: We corrected as suggested (line 144).

Line 115: Since root data are not available, it would be more accurate to say "aboveground productivity" throughout the manuscript to make it clear that that's what is being examined here.

Authors' answer: We replaced it with "aboveground organs". Further, at the end of this same paragraph we added a sentence just to ensure the reader has in mind we only used aboveground biomass in our analysis, reading:

"Therefore, we emphasize that in our study, forest biomass production and productivity were measured considering only above-ground tree components." (lines 157-158).

Besides, throughout the manuscript we replaced all terms "whole-plant" and "whole-plant elementomes" (previously used) by "aboveground organs" and "aboveground elementomes", respectively.

Line 118: It might be worth it to set the equations out in a separate line rather than having them embedded within the paragraph. That can make some of the math a little easier to follow.
Authors' answer: Yes, we set it in a separate line (151) as suggested.

Lines 119-120: here and throughout I found the use of production vs. productivity somewhat confusing. I understand the difference, and these may be the established conventions in which case this can be disregarded, but if it would be possible to rename one of the terms so they are more distinct, it would make it easier to follow which is being discussed later in the paper.
Authors' answer: We added new sentences at the fourth paragraph (lines 57-65) of the introduction section for ensuring the clarity between the two terms, reading:

"Considering elements (concentrations and stocks) of the entire aboveground biomass and leaves only may contribute to enhancing the understanding of ecosystem processes (Luo et al., 2020; Rocha et al., 2011). Forest biomass production (i.e., the overall total amount of biomass accumulated over an area in a given period) is influenced by the concentration of elements the plants store (Dar and Parthasarathy, 2022; Ullah et al., 2024). Fine roots, for example, influence tree nutrient stocks since they regulate processes like water absorption and nutrient uptake from the soil (Likulunga. et al., 2022; Zhao et al., 2022). Further, tree elemental concentrations (e.g., from aboveground organs) significantly impact ecosystem productivity (Bitomský et al., 2023; Elser et al., 2010). Therefore, elemental concentrations and stocks also contribute to forest biomass productivity—a unit of biomass (e.g., per area and year) produced per unit of standing biomass that reflects ecosystem efficiency (Margalef, 1998; Lartigue and Cebrian, 2012)."

Line 141: It would be a little clearer to put 'generalized additive mixed models' first and then put (GAMMs) in parentheses.
Authors' answer: We rephrased as suggested (line 177). Thank you.

Results

Line 191: Here 'forest functioning' is used a little too broadly. I'd suggest using productivity or production since that is what was measured.
Authors' answer: We replaced "forest functioning" with "forest biomass production and productivity".

Line 191-193: I found these sentences hard to follow. It seems like there are 2 models presented, one in each sentence, but they are each described as the 'best model.' Is the second sentence for productivity and that word has been omitted from the sentence? Authors' answer: For better clarifying it, we rephrased the second sentence of these two, now both read (lines 224-227):

"We found that the best model of forest biomass production using leaf element stocks as predictors explained 58% of the variance and had nine variables: C, Ca, K, Mg, N, P, C×N, C×P, and N×P (Fig. 1a). The second-best model explained 28% of the variance of forest biomass production (Fig. 1a) and had three aboveground element stocks as predictors (C, N, and C×N)."

Line 200: Nothing is mentioned about climate or age in the first paragraph but figure 1 seems to suggest that climate/age explains production best so might be worth mentioning that in this first paragraph.
Authors' answer: We added a new sentence for mentioning it. Further, we rephrased the other two sentences and merged into one for simplicity and clarity. These changes now read (lines 230-233):
"Forest biomass productivity was best predicted by the model with climate and stand age as predictors (Fig. 1c, d). Secondarily, between leaf elementomes (Ca, K, and N) and aboveground elementomes (K), the first ones were the best predictors of forest biomass productivity (Fig. 1c; 28% of variance explained)."

Discussion

Line 289-294: This may just be a wording issue, but it seems like these sentences are contradictory. The sentence starting "We found a possible effect…" suggests that less moisture caused plants to retain more nutrients in leaves to cope with drought. The sentence starting in line 293, "Therefore, our observed…" suggests that high precipitation coincides with high foliar nutrient storage, so it is unclear whether more nutrients are stored with more or less water.
Authors' answer: we rephrased the last sentence for adding more clarity. Now this part of the paragraph reads (lines 325-332):

"We found a positive effect of precipitation in the driest quarters, N and P, on forest biomass production. Since, during summer, most of the territory addressed in this study coincides with high temperatures and marked water stress (Martín Vide et al., 2008), plants may invest in a strategy of retaining larger foliar nutrient reserves to cope with drought (Waring, 1987.; Gessler et al., 2017). Increased precipitation might enhance the foliar nutrients stored in drier periods, thus contributing positively to aboveground biomass production (Fernández-Martínez et al., 2017; Lie et al., 2018; Roa-Fuentes et al., 2012). In our study region, high water availability (e.g., precipitation) correlates positively with mineralization, which enhances the nutrient availability to trees and contributes to increasing their biomass (Sardans et al., 2008)."

Line 325: It would be nice to restate more specifically here what the decrease in forest biomass is that the authors refer to.
Authors' answer: We rephrased it for clarity, reading (lines 361-364):

"Lastly, the lower relevance of C in our average models may be partially due to its variations across distinct plant organs, e.g., the predominance of leaf and fine-root turnovers in C allocations (Yu et al., 2017). Besides, foliar nutrients, particularly P, significantly impact photosynthetic C uptake in forests, promoting variation in biomass production (Mercado et al., 2011)."

Conclusion

Lines 343-345: The authors point out that productivity was not driven by nutrients but stand age, then suggest that focusing on leaf elements is sufficient for understanding variations in forest biomass. This seems a little misleading–more weight should be given to the fact that other variables besides leaf nutrients were primary drivers and that root and soil nutrients were not considered as a part of this analysis.
Authors' answer: we toned down the statement by rephrasing the last sentence, now reading:

"Altogether, our results indicate that leaf element stocks are critical predictors of forest biomass production" (lines 409-410).

Regarding the recognition of the role of other factors like soil and roots, we already addressed it the section "Caveats, limitations, and implications" (lines 374-399), described in answers to previous comments.

---

## Author Response (AR2)

**Answer to editor**

Dear authors,

Thank you for the thorough work revising this manuscript. Both reviewers are satisfied with the changes, and agree that this manuscript would be a nice addition to the literature. One of the reviewers mentions - however - that there are still some typo's in the current version. Could you revise thoroughly for these small errors, and then resubmit please?

Thanks in advance,

Marijn

Authors' answer: Dear editor, we are glad that the previous revisions fulfill the quality expected by your journal. We revised carefully the manuscript searching for typos or other minor errors and corrected them accordingly. Further, in the answers to the referee 2, we cite the lines where these corrections were made. If there are any further adjustments necessary to make, please just let us know. Thank you for your interest in our work.

**Answer to reviewer #1: Vallicrosa, Helena:**

I am satisfied with the effort made by the authors in addressing the raised concerns in the previous round of reviews. I believe that this contribution is now suitable for publication as is.

Authors' answer: Dear referee, thank you very much for your contribution to the previous versions of the manuscript. Your revision contributed to the improvement of our work.

**Answer to reviewer #2: Hauser, Emma:**

The authors have nicely revised this paper to improve language regarding the study's scope and implications. The paper now highlights that this study is specific to forests and aboveground productivity, which ultimately makes the authors conclusions stronger. I especially appreciated revisions to the introduction to include a discussion of soil nutrients and a more nuanced conversation surrounding elementomes and the role of C versus nutrients in plant growth. The introduction was much more streamlined and sets the stage for the results very nicely.

It was also great to see the authors discussion of root and soil variables in the discussion/caveats section. This would be a nice step for future work and it's helpful to have this included to keep the conversation going in the research community.

There were a few remaining typos, so it would be good to do a technical double check before the final publication, but I have no other substantive suggestions to make. This manuscript makes a nice contribution to our understanding of aboveground elemental cycling, its interaction with environmental variables, and its predictive capacity for forest productivity.

Authors' answer: Dear referee, thank you for your feedback on our previous revision. We also acknowledge your detailed review that allowed us to improve the points and topics you mentioned. We have carefully reviewed the manuscript and corrected any typos or

minor errors and made minor and minor adjustments to verb agreement. Below we cite the lines where these corrections were made.

Lines:
L29 (deleted the duplicate of the word 'drive')
L41 ('45% in bark and woods' replaced by ('45% in bark and wood')
L71 ('effects of the tree nutrient stocks' replaced by ('effects of tree nutrient stocks')
L147 ('...tree diameters' replaced by '....data on tree diameter')
L149 ('access' replaced by 'see in detail')
L160 ('was' replaced by 'were')
L163 (added 'values of the stocks')
L188 ('smoothed' replaced by 'smooth')
L189-190 (added the word 'functions' to the follow the term 'spline (Edf)')
L197 ('elementome' replaced by 'elementomes')
L232 (added the proper parenthesis to cite Fig. 1a).
L272 ('...were unable to predict...' replaced by '...did not significantly predict...')
L289 (Figure 4, deleted '* Indicates significant coefficient')
L297 (Figure 5, deleted '* Indicates significant coefficient')
L311 ('stocks aboveground' replaced by 'aboveground stocks')
L385 ('above ground' replaced by 'aboveground')
L390 ('below-ground' replaced by 'belowground')
L410-411 ('...were the most positively correlated...' replaced by '...showed the strongest positive correlation...')